# DualFocus: Depth from Focus with Spatio-Focal Dual Variational Constraints

**Sungmin Woo**
Yonsei University
smw3250@yonsei.ac.kr

**Sangyoun Lee**
Yonsei University
syleee@yonsei.ac.kr

## Abstract

Depth-from-Focus (DFF) enables precise depth estimation by analyzing focus cues across a stack of images captured at varying focal lengths. While recent learning-based approaches have advanced this field, they often struggle in complex scenes with fine textures or abrupt depth changes, where focus cues may become ambiguous or misleading. We present DualFocus, a novel DFF framework that leverages the focal stack's unique gradient patterns induced by focus variation, jointly modeling focus changes over spatial and focal dimensions. Our approach introduces a variational formulation with dual constraints tailored to DFF: spatial constraints exploit gradient pattern changes across focus levels to distinguish true depth edges from texture artifacts, while focal constraints enforce unimodal, monotonic focus probabilities aligned with physical focus behavior. These inductive biases improve robustness and accuracy in challenging regions. Comprehensive experiments on four public datasets demonstrate that DualFocus consistently outperforms state-of-the-art methods in both depth accuracy and perceptual quality.

## 1   Introduction

Depth estimation plays a pivotal role in 3D vision, enabling a wide range of applications from 3D reconstruction to augmented reality and robotics. Among various techniques available, Depth-from-Focus (DFF) offers a passive, hardware-free approach that utilizes a focal stack—a series of images captured at different focal distances. This approach is particularly attractive for consumer-grade imaging systems, such as smartphone cameras, which often lack specialized stereo or depth sensors and must infer depth from limited inputs like a single image or a focal sweep.

Unlike monocular or stereo-based depth estimation, DFF is grounded in a simple yet robust physical principle: a scene point appears sharpest when the camera's focal plane aligns with its depth. This focus-sharpness relationship provides an interpretable signal that, with proper calibration, can yield accurate metric depth. Moreover, DFF is inherently free from scale ambiguity—an issue prevalent in monocular depth estimation [31, 19, 31], and does not require scene-specific priors or extensive supervision [3, 30]. Despite these strengths, classical DFF techniques [16, 18, 10, 1, 17, 28] suffer from critical limitations. They rely on heuristic focus measures (e.g., contrast or sharpness) and apply post-hoc smoothing to infer depth. As a result, these methods often falter in the presence of ambiguous focus responses, fine textures, or abrupt depth changes, producing depth maps that are either too blurry or riddled with noise.

Recent advances in learning-based DFF [26, 9, 6, 29] have tackled some of these issues by training deep networks to implicitly learn focus patterns from data. While these models achieve notable improvements in benchmark performance, they typically overlook the underlying physical and geometric principles, such as the gradual changes in sharpness driven by light's focus behavior in a camera. In particular, they do not fully exploit the complementary roles of spatial and focal information, which are essential for accurate and robust DFF in complex real-world scenes.

39th Conference on Neural Information Processing Systems (NeurIPS 2025).

In this paper, we propose *DualFocus*, a novel framework tailored for DFF, which jointly models focus variation across spatial and focal dimensions. Unlike conventional learning-based DFF methods that treat focal stack images similarly to all-in-focus RGB frames or depth maps, DualFocus introduces a physics-aware design that leverages the unique structure of focus cues inherent in focal stacks.

At the core of our approach are two variational constraints grounded in the optical principles of focus. First, the spatial variational constraint focuses not on estimating sharpness directly, but on analyzing the spatial gradient patterns that emerge across focal planes. Since each focal slice captures a different focus distance, the same scene point exhibits varying gradient patterns depending on whether it is in or out of focus. In-focus regions tend to show coherent, strong gradients, while out-of-focus regions exhibit diffused or noisy patterns. By comparing these spatial gradient patterns across the stack, the model learns to infer sharpness indirectly and identify reliable depth edges—discriminating them from texture-induced gradients, whose patterns differ from the focus-dependent changes observed at true depth edges. Second, the focal variational constraint leverages the continuous and ordered nature of focal transitions in a focal stack. At each spatial location, it encourages a unimodal and bidirectionally monotonic distribution of focus probabilities along the focal axis, ensuring that the predicted focus confidence peaks at the in-focus plane and decreases smoothly as the focal distance diverges in either direction. This physically grounded constraint reflects how real-world points appear progressively blurrier when the focal distance moves away from their true depth, and allows the model to robustly disambiguate noisy focus cues, especially in textureless or reflective regions. Together, these constraints introduce domain-specific inductive priors that reflect the unique imaging characteristics of focal stacks, significantly improving robustness and generalization in diverse and challenging DFF scenarios.

Extensive experiments on four public datasets, including NYU Depth v2 [25], FoD500 [15], DDFF 12-Scene [9], and ARKitScenes [2], demonstrate that DualFocus surpasses state-of-the-art methods in both depth accuracy and perceptual quality.

Our main contributions are:

- A novel variational approach to DFF that leverages spatial and focal focus variations through constraints grounded in optical principles.

- Two tailored variational constraints: (i) spatial constraints that analyze gradient pattern changes across focal planes to identify reliable depth edges, and (ii) focal constraints that enforce unimodal, monotonic focus probabilities aligned with physical focus behavior.

- A unified differentiable architecture that integrates these constraints into an end-to-end optimization framework, enabling seamless training with standard depth supervision.

- State-of-the-art performance on multiple benchmarks (NYU Depth v2, FoD500, DDFF 12-Scene) and impressive generalization to unseen dataset (ARKitScenes).

## 2 Related Work

**Depth-from-Focus.** DFF, also known as shape-from-focus, exploits the principle that objects within a camera's depth of field (DoF) appear sharp, while those outside form a blurred circle of confusion (CoC) due to lens effects [23, 20]. Early DFF techniques analyze a focal stack to find the frame where each pixel is maximally sharp, using that focal distance as a proxy for depth [20, 16]. Sharpness is typically measured using Laplacian-based or frequency-domain operators [16, 15], but these methods often require densely sampled stacks and degrade in low-texture regions due to noise sensitivity [20].

The advent of deep learning has transformed DFF by introducing data-driven solutions. DDFFNet [9] introduced an end-to-end CNN trained on the DDFF 12-Scene dataset, while AiFNet [22] combined supervised and unsupervised learning to operate with or without ground-truth depth. DefocusNet [15] exploits the CoC-based defocus cue to produce intermediate defocus maps. DEReD [24] adopts a self-supervised approach that reconstructs both depth and all-in-focus images from focal stacks via learned optical defocus simulation. DFVNet [29] captures first-order derivatives of volumetric features across focal planes to guide depth estimation. HybridDepth [6] leverages pretrained relative depth models and refines them into metric depth using a DFF backbone.

While these approaches effectively process focal differences and sharpness cues, they typically infer depth directly from appearance features, vulnerable to texture-induced artifacts. In contrast, our

DualFocus introduces spatial variational constraints that leverage focus-dependent gradient variations to capture additional focus cues, and focal variational constraints that enforce unimodal, monotonic focus probability distributions aligned with physical focus transitions. This dual approach, unique to DFF, enhances accuracy and robustness in complex scenes with ambiguous focus patterns.

**Spatial variation modeling in depth estimation.** Modeling spatial variation is a cornerstone of depth estimation, enabling the capture of geometric structure through relationships between neighboring pixels. Several methods have leveraged gradient-based techniques or spatial priors to enhance depth prediction accuracy. Some studies [21, 4] focus on iteratively refining depth maps from off-the-shelf networks. An affinity matrix is introduced to learn pixel-neighbor depth relationships, though its unsupervised nature limits precision [5]. Li et al. [12] explore fusing gradients and depth either through unconstrained end-to-end networks or non-differentiable optimization. More recently, VA-DepthNet [14] uses first-order variational constraints to depth gradients in single-image depth estimation, reconstructing depth from a gradient-aware surface field using least-squares optimization.

These techniques demonstrate the benefits of spatial regularization, yet they operate in single-image or general depth settings without access to DFF's rich multi-focus information. Our method uniquely adapts spatial variational constraints to DFF by leveraging sharpness variation across focal planes, which enables the model to distinguish true depth edges from spurious textures by contrasting gradient responses across focus levels. In this paper, we demonstrate how spatial priors from variational formulations can be reinterpreted to benefit focus-based depth estimation.

## 3 Method

### 3.1 Overview

Our method estimates depth from a focal stack by modeling the feature variations induced by focus changes and imposing variational constraints in both spatial and focal domains. As described in Section 3.2, we first construct a 4D focus volume from the focal stack to capture discriminative representations across spatial and focus dimensions. We then introduce two complementary variational constraints: (1) **spatial variational constraints**, which exploit focus-induced gradient variation to reconstruct reliable, integrable depth gradients while suppressing texture-driven noise (Section 3.3), and (2) **focal variational constraints**, which encourage unimodal focus probability distributions that align with physical focus behavior (Section 3.4). The overview of our method is depicted in Figure 1.

### 3.2 Focus Volume Modeling

Given a focal stack of $N$ images captured at distinct focal distances, we begin by independently extracting feature maps from each image. These feature maps are then stacked along the focal dimension to form a 4D focus volume $V \in \mathbb{R}^{H \times W \times C_1 \times N}$, where $H$, $W$, $C_1$ denote the height, width, and the number of channels, respectively. To maintain consistency in depth interpretation, the focal stack is ordered by increasing focal distance. Similar to DFV [29], we compute differences along focal dimension, which captures the variation of image features across different focal settings. We concatenate these features with the original feature map in the channel dimensions to obtain the augmented focus volume $V^* \in \mathbb{R}^{H \times W \times 2C_1 \times N}$, serving as the basis for subsequent focus analysis:

$$V_n^* = \begin{cases} [V_n, V_{n+1} - V_n], & n = 1, \ldots, N-1 \\ [V_n, V_n - V_{n-1}], & n = N \end{cases} \tag{1}$$

where $[:,:]$ represents concatenation operation.

### 3.3 Spatial Variational Constraints

Rather than directly regressing absolute depths, our network predicts the first-order differences between neighboring pixels, representing local depth gradients that capture how depth varies across the scene. While such gradients have often been used as regularization terms, we adopt them as primary prediction targets within the context of focal-stack depth estimation. This allows the network to encode focus-dependent structural variations that are imperceptible to all-in-focus approaches.

In conventional single-image depth estimation (SIDE), gradients are extracted uniformly across the image regardless of whether they arise from true depth edges or from irrelevant textures. This

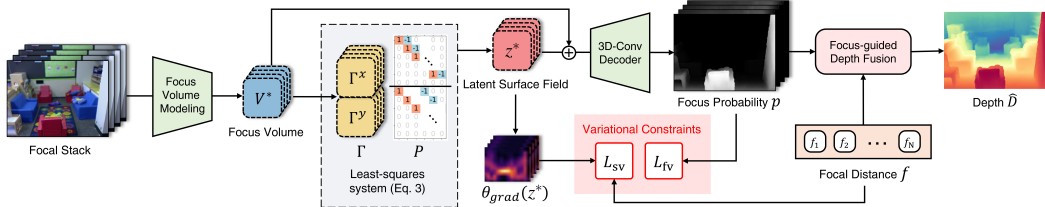

Figure 1: Overview of the proposed DualFocus. Predicted spatial gradients $\Gamma = (\Gamma^x, \Gamma^y)$ are fitted to an implicit surface $z^*$ via a least-squares system, where sharp, integrable gradients reconstruct coherent surfaces in in-focus regions of focal stack data. The focus probability $p$ is constrained to a smooth, unimodal distribution by $L_{\text{fv}}$, reflecting the gradual nature of focus shifts across the stack.

indiscriminate extraction induces ambiguity—especially in repeating-pattern regions—where strong gradients do not necessarily indicate depth discontinuities. Crucially, SIDE methods lack a mechanism to distinguish reliable gradients from noisy ones. Focal stacks, by contrast, inherently encode focus-based variation correlated with depth: a pixel may appear sharp (strong, reliable gradients) in one plane and blurred (weak, noisy gradients) in another. By comparing gradient predictions across $N$ focal planes, our model learns to discern reliable depth cues from spurious texture signals.

For each focal plane $n$, we predict a multi-channel gradient feature $\Gamma_n$:

$$\Gamma_n = \left[ \Gamma_n^{(1)}, \ldots, \Gamma_n^{(C_2)} \right] \in \mathbb{R}^{2HW \times C_2}, \tag{2}$$

where each channel $\Gamma_n^{(c)} \in \mathbb{R}^{2HW}$ encodes learned x-axis and y-axis depth-variation cues at a relatively low spatial resolution ($14 \times 14$ in our model), capturing coarse, semantically meaningful focus-induced structural patterns. As each focal plane emphasizes different depth layers, the resulting $\Gamma_n$ fields vary significantly across $n$, reflecting the "focus dependence" we aim to leverage.

However, direct supervision of $\Gamma_n$ as depth gradients leads to noisy, spatially inconsistent results, as shown in Figure 2. This arises from the absence of global integrability constraints, producing gradient patterns that no real surface could yield. To regularize this, we project $\Gamma_n$ onto the space of integrable (curl-free) gradient fields by solving a least-squares optimization. This reconstructs the closest scalar surface whose gradient approximates $\Gamma_n$ in the least-squares sense. Our formulation is inspired by VA-DepthNet [14], which applies it to exploit surface field as an intermediate depth representation in a single all-in-focus image. In contrast, we extend the concept to focal stacks, enabling focus-dependent surface representations that better capture variations in geometric consistency across focal planes.

Specifically, we solve the overdetermined system separately for each channel $c$:

$$P z_n^{*(c)} = \Gamma_n^{(c)}, \quad z_n^{*(c)} = \arg\min_z \| P z - \Gamma_n^{(c)} \|_2^2 = (P^\top P)^{-1} P^\top \Gamma_n^{(c)}, \tag{3}$$

where $P \in \{-1, 0, 1\}^{2HW \times HW}$ is a fixed finite-difference operator (e.g., Sobel-style stencil) that computes horizontal and vertical derivatives for each pixel. This operator maps a scalar field (e.g., depth) to its spatial gradient, and solving the least-squares system inverts this mapping—yielding $z_n^* \in \mathbb{R}^{HW \times C_2}$, a reconstructed surface representation whose spatial gradient best matches the predicted $\Gamma_n$ in the least-squares sense. The derivation of the optimal solution in Eq. 3 is provided in Appendix B.

Since $\Gamma_n$ reflects the gradient of an image focused at focal distance $f_n$, the reconstructed $z_n^*$ naturally varies across $n$. In-focus planes tend to yield coherent structures consistent with scene geometry, while out-of-focus planes produce noisy, non-integrable surfaces. This discrepancy implicitly encodes the reliability of geometric cues per focal settings. Importantly, this property introduces a beneficial inductive bias during training that only in-focus regions can be consistently reconstructed into plausible surfaces, prompting the network to learn $\Gamma_n$ fields whose gradient energy is concentrated in those regions. As a result, without explicit supervision for focus-awareness, the model learns to produce $\Gamma_n$ representations that are inherently focus-sensitive. Figure 2 illustrates the effect: integrability-regularized gradients highlight focused areas more effectively than directly supervised counterparts.

To further guide this behavior, we supervise $z_n^*$ only where the focal plane is actually in focus. For a given pixel $\mathbf{x}$, we define a per-pixel, per-plane sharpness weight $q_n$ indicating how close the plane

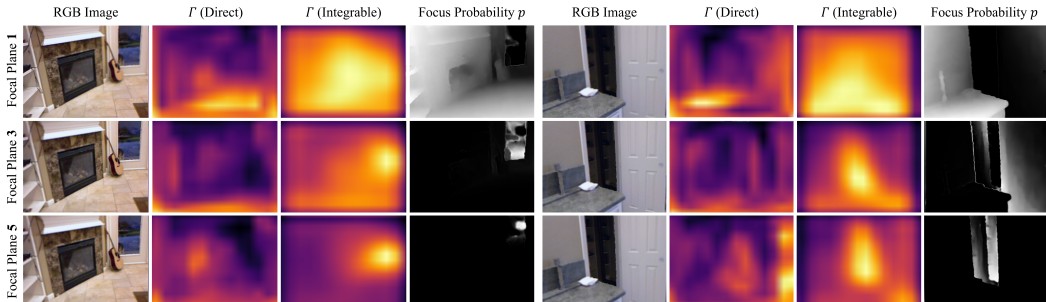

Figure 2: Effect of integrability-based reconstruction on gradient fields $\Gamma = (\Gamma^x, \Gamma^y)$ with $N = 5$. Each $\Gamma^x, \Gamma^y \in \mathbb{R}^{14 \times 14 \times C_2}$ is upsampled and normalized, highlighting focus sensitivity.

with focal distance $f_n$ is to the ground-truth depth $D^*(\mathbf{x})$:

$$q_n(\mathbf{x}) = \frac{\exp\big(-|f_n - D^*(\mathbf{x})|\big)}{\sum_{m=1}^{N} \exp\big(-|f_m - D^*(\mathbf{x})|\big)}. \tag{4}$$

This weight peaks at the plane closest to the true focus depth. Using $q(\mathbf{x})$, we define the spatial variational loss as:

$$L_{\text{sv}} = \sum_{\mathbf{x},n} q_n(\mathbf{x}) \left\| \nabla D^*(\mathbf{x}) - \theta_{\text{grad}}\big(z_n^*\big)(\mathbf{x}) \right\|_1, \tag{5}$$

where $\nabla$ denotes the gradient operator, and $\theta_{\text{grad}}$ is a $3 \times 3$ convolution that fuses the $C_2$ channels of $z_n^*$ into a 2-channel gradient prediction for the horizontal and vertical directions. By weighting with $q_n(\mathbf{x})$, the loss encourages alignment with true surface geometry only in sharp, in-focus regions—thus avoiding overfitting to unreliable noisy gradients in defocused areas. Visualizations of the surface field gradients are provided in Appendix C.3.

**Focus-guided depth fusion.** The reconstructed implicit surface feature $z^* \in \mathbb{R}^{H \times W \times C_2 \times N}$ is then concatenated with the focus volume $V^* \in \mathbb{R}^{H \times W \times 2C_1 \times N}$, resulting in a fused feature of shape $H \times W \times (2C_1 + C_2) \times N$. The fused feature is processed by a cascade of 3D convolutional layers with upsampling, which jointly reason across spatial and focal dimensions. The decoder outputs focus probability maps $p \in \mathbb{R}^{H' \times W' \times N}$ representing sharpness, with $H' \times W'$ matching the resolution of the input image, and are normalized via softmax such that $\sum_{n=1}^{N} p_n(\mathbf{x}) = 1$. Finally, the depth $\widehat{D}(\mathbf{x})$ is estimated by computing the weighted sum of the known focal distances $\{f_n\}$ using the predicted probabilities $p_n$ as weights:

$$\widehat{D}(\mathbf{x}) = \sum_{n=1}^{N} p_n(\mathbf{x}) f_n. \tag{6}$$

In essence, by reconstructing multi-channel focus-dependent surface features $z_n^*$ and supervising only their in-focus consistency, we extract structural cues unavailable to single-image methods. The differences among the $z_n^*$ across focal distances serve as the key signals that our 3D-Conv decoder learns to fuse, leading to sharply localized depth discontinuities and robust absolute depth estimation.

### 3.4 Focal Variational Constraints

In DFF, the continuity of focus transitions across focal planes reflects the physical behavior of light and optics. To exploit this inherent property, we introduce variational constraints that enforce smooth and unimodal focus probability distributions along the focal axis. Unlike prior methods that treat focal slices independently or apply limited regularization, our formulation explicitly models inter-plane consistency, enabling coherent and physically plausible depth estimation—especially in complex or low-texture regions.

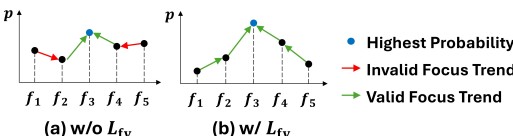

Figure 3: Our focal variational constraints ensure plausible focus probability distribution, reflecting natural focus transitions across the focal stack.

To this end, we define a focal variation loss that promotes unimodality and monotonicity in the per-pixel focus probabilities across the focal stack. Specifically, we introduce a bidirectional soft

monotonicity loss:

$$L_{\text{fv}} = \sum_{\mathbf{x}} \left( \sum_{i=1}^{k(\mathbf{x})-1} \left( \max(0, p_i(\mathbf{x}) - p_{i+1}(\mathbf{x})) \right)^2 + \sum_{i=k(\mathbf{x})}^{N-1} \left( \max(0, p_{i+1}(\mathbf{x}) - p_i(\mathbf{x})) \right)^2 \right), \quad (7)$$

where $k(\mathbf{x}) = \arg\max_n p_n(\mathbf{x})$ denotes the index of the focal plane with the highest focus probability at pixel $\mathbf{x}$. The first summation penalizes decreases before the peak (encouraging a rising slope), while the second penalizes increases after the peak (encouraging a falling slope). Together, they softly enforce a unimodal distribution centered at the most probable focal plane.

This loss aligns with physical intuition: focus probability should peak at the correct depth and diminish smoothly as the focal plane deviates. By integrating $L_{\text{fv}}$ into training, the model learns to produce spatially and temporally coherent focus responses, which in turn improve depth accuracy and robustness across diverse focal conditions. Further analysis of the effect of the focal variational constraint is provided in Appendix C.4.

### 3.5 Loss Function

The overall training objective integrates three complementary loss terms designed to guide both spatial and focal reasoning:

$$L = L_{\text{depth}} + \lambda_{\text{sv}} L_{\text{sv}} + \lambda_{\text{fv}} L_{\text{fv}}, \quad (8)$$

where $L_{\text{depth}}$ is the smooth L1 loss between the ground-truth $D^*$ and the predicted depth $\widehat{D}$, and $\lambda_{\text{sv}}$ and $\lambda_{\text{fv}}$ are balancing scalars. By combining these three objectives, our approach leverages the dual variational constraints to robustly infer fine-grained depth from focal stacks. The spatial term ensures focus-dependent latent surface cues, while the focal term preserves global focus dynamics—jointly enabling precise depth localization and smooth transitions across scenes.

## 4 Experiments

### 4.1 Experimental Setup

**Datasets.** To evaluate our proposed method, we leverage four distinct datasets, each offering unique characteristics that enable assessment across diverse scenarios ranging from synthetic to real-world conditions. For FoD500 and DDFF 12-Scene datasets, we follow the training protocol used in DFV [29]. **(1) NYU Depth v2** [25] is a comprehensive indoor dataset with over 24,000 RGB-depth pairs for training and 654 for testing. Since it lacks native focal stack images, we generate synthetic focal stacks using the dataset synthesis technique from HybridDepth [6], adapting it for our DFF framework. The procedure for generating the synthetic focal stacks is described in the Appendix A.2. For single-image depth estimation (SIDE) baselines, we use the all-in-focus RGB images rather than any blurred images from the focal stack, as SIDE models are generally trained on all-in-focus data. **(2) FoD500** [15] is a synthetic dataset originally designed for DFD, featuring 400 training and 100 test samples. Each includes a 5-frame focal stack and a ground truth depth map. The image resolution is $256 \times 256$, which is randomly cropped into $224 \times 224$. **(3) DDFF 12-Scene** [9] is a real-world dataset tailored for DFF evaluation, captured via a light-field camera across 12 scenes. We adopt the split from DFV [29], using six scenes for training and validation (e.g., kitchen, seminaroom) and six for testing (e.g., cafeteria, library). Each sample provides a 10-frame focal stack, though we use randomly selected 5 frames for consistency. Training uses $224 \times 224$ random crops and flips, while evaluation is performed at the original resolution of $383 \times 552$, consistent with prior works [29, 6, 9]. **(4) ARKitScenes** [2] is a large-scale mobile AR dataset. We use a subset of 5,600 images for zero-shot evaluation to assess the model's ability to generalize to unseen real-world environments without fine-tuning.

**Metrics.** We evaluate our method's performance using a suite of standard depth estimation metrics: Mean Squared Error (MSE), Root Mean Squared Error (RMSE), Absolute Relative Error (AbsRel), Squared Relative Error (SqRel), Accuracy ($\delta_1$, $\delta_2$, $\delta_3$), and Bumpiness (Bump). Detailed formulas for each metric are provided in the Appendix A.4.

**Implementation details.** We train our model on two Titan RTX GPUs using PyTorch. The encoder is based on a ResNet-18 FPN [13] and the decoder employs 3D-ResNet blocks [8]. For optimization,

Table 1: Performance comparison on the NYU Depth V2 dataset with single-image depth estimation (SIDE) and depth from focus/defocus (DFF/DFD) methods. † ZoeDepth-M12-N model is used for evaluation. ‡ indicates results manually reproduced using the released code.

| Model | Type | RMSE ↓ | AbsRel ↓ | $\delta_1$ ↑ | $\delta_2$ ↑ | $\delta_3$ ↑ |
|---|---|---|---|---|---|---|
| ZoeDepth[†] [3] | SIDE | 0.270 | 0.075 | 0.955 | 0.995 | 0.999 |
| VPD [31] | SIDE | 0.254 | 0.069 | 0.964 | 0.995 | 0.999 |
| Marigold [11] | SIDE | 0.224 | 0.055 | 0.964 | 0.991 | 0.998 |
| ECoDepth [19] | SIDE | 0.218 | 0.059 | 0.978 | 0.997 | 0.999 |
| Depth Anything [30] | SIDE | 0.206 | 0.056 | 0.984 | 0.998 | **1.000** |
| DefocusNet [15] | DFD | 0.493 | - | - | - | - |
| HybridDepth [6] | DFF | 0.128 | 0.026 | 0.995 | **1.000** | **1.000** |
| DFV[‡] [29] | DFF | 0.094 | 0.020 | 0.998 | **1.000** | **1.000** |
| **Ours** | DFF | **0.075** | **0.013** | **0.999** | **1.000** | **1.000** |

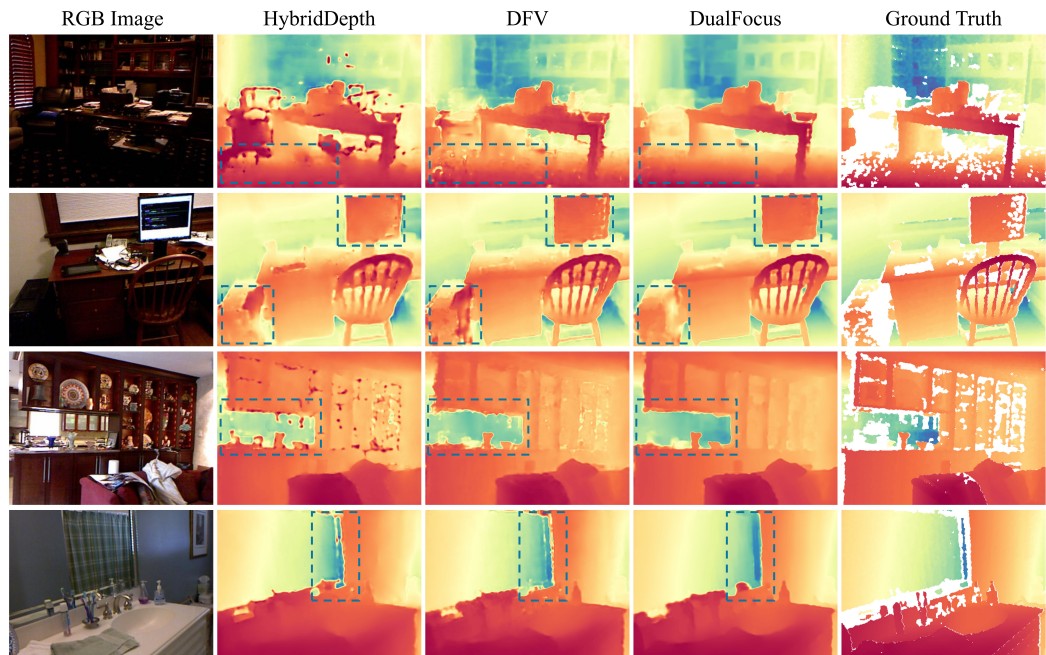

Figure 4: Qualitative comparisons on the NYU Depth v2 dataset.

we use the Adam optimizer ($\beta 1 = 0.9$, $\beta 2 = 0.999$) with an initial learning rate of $1 \times 10^{-4}$, which is reduced to $1 \times 10^{-5}$ via a cosine annealing scheduler. The model is trained for 40 epochs on the NYU Depth v2 dataset with a batch size of 16, and for 2000 epochs on the FoD500 and DDFF 12-Scene datasets with a batch size of 20. To estimate the latent surface field $z^*$, we solve the regularized normal equation in closed form using *torch.linalg.solve* as in [14], which is efficiently executed on the GPU. Since this computation is performed at a coarse feature resolution ($14 \times 14$ pixels), it introduces minimal overhead and enables fast, stable inference without requiring iterative optimization. Further implementation details are provided in Appendix A.1, and a detailed description of the model architecture is given in Appendix A.3.

## 4.2 Comparison with the State-of-the-Art

**NYU Depth v2 Dataset.** We evaluate our DualFocus on the NYU Depth v2 dataset, a widely adopted benchmark for indoor depth estimation. As summarized in Table 1, DualFocus consistently outperforms both single-image depth estimation (SIDE) and DFF/DFD baselines. Compared to the previous state-of-the-art DFV [29], DualFocus achieves a 20.2% reduction in RMSE (from 0.094 to 0.075) and a 35.0% reduction in AbsRel (from 0.020 to 0.013), while also improving $\delta_1$. Figure 4

Table 2: Performance comparison on the FoD500 dataset.

| Model | MSE ↓ | RMSE ↓ | AbsRel ↓ | SqRel ↓ | $\delta_1$ ↑ | $\delta_2$ ↑ | $\delta_3$ ↑ | Bump ↓ |
|---|---|---|---|---|---|---|---|---|
| RDF [10] | 0.112 | 0.322 | 0.46 | 0.240 | 0.395 | 0.646 | 0.761 | 1.54 |
| DDFF [9] | 0.033 | 0.167 | 0.17 | 0.036 | 0.728 | 0.900 | 0.963 | 1.74 |
| Defocus-Net [15] | 0.022 | 0.134 | 0.15 | 0.036 | 0.811 | 0.933 | 0.966 | 2.52 |
| DFV [29] | 0.020 | 0.129 | **0.13** | 0.024 | 0.819 | 0.947 | **0.980** | 1.43 |
| **Ours** | **0.015** | **0.112** | **0.13** | **0.022** | **0.829** | **0.948** | **0.980** | **1.31** |

Table 3: Performance comparison on the DDFF 12-Scene dataset.

| Model | MSE ↓ | RMSE ↓ | AbsRel ↓ | SqRel ↓ | $\delta_1$ ↑ | $\delta_2$ ↑ | $\delta_3$ ↑ | Bump ↓ |
|---|---|---|---|---|---|---|---|---|
| RDF [10] | $91.8 \times 10^{-4}$ | 0.0941 | 1.00 | 0.1394 | 0.156 | 0.331 | 0.475 | 1.33 |
| DDFF [9] | $8.9 \times 10^{-4}$ | 0.0276 | 0.24 | 0.0095 | 0.613 | 0.887 | 0.965 | 0.52 |
| Defocus-Net [15] | $8.6 \times 10^{-4}$ | 0.0255 | 0.17 | 0.0060 | 0.726 | 0.942 | 0.979 | 0.46 |
| DFV [29] | $5.7 \times 10^{-4}$ | 0.0213 | 0.17 | 0.0063 | 0.767 | 0.942 | 0.981 | 0.42 |
| HybridDepth [6] | $5.1 \times 10^{-4}$ | 0.0200 | 0.17 | 0.0060 | 0.789 | 0.947 | 0.981 | 0.47 |
| **Ours** | **$4.7 \times 10^{-4}$** | **0.0194** | **0.16** | **0.0056** | **0.800** | **0.954** | **0.982** | **0.40** |

Table 4: Zero-shot evaluation comparison on the ARKitScenes validation set with SIDE and DFF methods. † ZoeDepth-M12-N model is used for evaluation.

| Model | Type | RMSE ↓ | AbsRel ↓ | #Params |
|---|---|---|---|---|
| ZoeDepth[†] [3] | SIDE | 0.61 | 0.33 | 335M |
| DistDepth [27] | SIDE | 0.94 | 0.45 | 69M |
| ZeroDepth [7] | SIDE | 0.62 | 0.37 | 233M |
| Depth Anything [30] | SIDE | **0.53** | **0.32** | 336M |
| DFV [29] | DFF | 0.43 | 0.51 | 20M |
| HybridDepth [6] | DFF | 0.29 | 0.42 | 67M |
| **Ours** | DFF | **0.28** | **0.40** | 27M |

further illustrates DualFocus's qualitative superiority over DFV and HybridDepth, demonstrating robustness to repeating floor patterns, consistent depth within objects, and enhanced edge preservation. This substantial improvement across quantitative and qualitative metrics demonstrates the superiority of our focal-aware design, particularly in leveraging variational focus cues for more accurate depth estimation in indoor environments.

**FoD500 Dataset.** The FoD500 dataset contains scenes with intricate structures and frequent depth discontinuities, making it particularly challenging for DFF approaches. As presented in Table 2, DualFocus achieves the best performance across all evaluation metrics. In particular, it achieves a Bump score of 1.31, outperforming the previous best (DFV: 1.43) by 8.4%. These results emphasize our model's capability to maintain sharp depth boundaries and effectively leverage both spatial and focal gradients, leading to perceptually faithful depth reconstructions.

**DDFF 12-Scene Dataset.** The DDFF 12-Scene dataset includes diverse focal stack conditions, providing a rigorous testbed for evaluating DFF methods. Table 3 shows that DualFocus achieves state-of-the-art performance across all metrics. It yields an MSE of $4.7 \times 10^{-4}$ and RMSE of 0.0194, marking a notable improvement over prior methods such as HybridDepth (MSE: $5.1 \times 10^{-4}$) and DFV (MSE: $5.7 \times 10^{-4}$). The enhanced accuracy highlights our model's robustness in resolving depth ambiguities under varying focus cues, supported by spatially-aware focal supervision.

## 4.3 Zero-Shot Transfer

To assess generalization, we perform zero-shot depth estimation on the ARKitScenes dataset. DFF models including ours are trained solely on the NYU Depth V2 dataset, while SIDE models leverage a broad ensemble of diverse datasets. As reported in Table 4, DualFocus outperforms other DFF baselines, including HybridDepth [6], which leverages pretrained Depth Anything [30] results for strong zero-shot transfer, and achieves performance competitive with SIDE methods. This highlights its robust generalization to unseen focal stacks, driven by the spatio-focal variational constraints.

Table 5: Ablation studies of the proposed components.

| Method | RMSE ↓ | log RMSE ↓ | AbsRel ↓ | SqRel ↓ |
|---|---|---|---|---|
| w/o spatio-focal variational constraints | 0.094 | 0.027 | 0.020 | 0.0038 |
| w/o spatial variational constraints | 0.090 | 0.025 | 0.018 | 0.0032 |
| w/o focal variational constraints | 0.078 | 0.022 | 0.014 | 0.0022 |
| w/ direct supervision on gradients $\Gamma$ | 0.083 | 0.023 | 0.015 | 0.0026 |
| w/o sharpness weight $q$ | 0.077 | 0.022 | 0.014 | 0.0022 |
| w/ blurness weight $(1-q)$ | 0.079 | 0.022 | 0.014 | 0.0025 |
| Ours | **0.075** | **0.020** | **0.013** | **0.0021** |

## 4.4 Ablation Study

We conduct comprehensive ablation studies on the NYU Depth v2 dataset to validate the effectiveness of the proposed spatio-focal variational constraints, as summarized in Table 5.

**Spatio-Focal Variational Constraints.** When both spatial and focal variational constraints are removed, the model's performance significantly degrades across all metrics (e.g., RMSE increases from 0.075 to 0.094), underscoring the importance of incorporating variational regularization in DFF. To further disentangle their individual contributions, we evaluate the effects of removing each constraint separately: Omitting the spatial variational constraint leads to a marked performance drop, validating the effectiveness of focus-dependent spatial gradients modeling. Removing the focal variational constraint also results in degraded performance, indicating its auxiliary role in guiding the focus probability distributions to reflect natural defocus behaviors. Overall, the spatial variational constraint demonstrates greater impact than its focal counterpart, as it directly encodes surface gradients at each focal plane, capturing localized geometric cues that vary with focus.

**Effect of Integrability-based Regularization on $\Gamma$.** To assess the effectiveness of our integrability-based regularization scheme, we compare it against a baseline that applies naive gradient-level supervision directly on the predicted gradient field $\Gamma$. Without enforcing integrability, this baseline yields consistently higher errors across all evaluation metrics, as unconstrained gradients yield spatially inconsistent or non-physical surfaces. In contrast, our integrability projection reconstructs curl-free surface approximations from $\Gamma$, regularizing the gradient field toward geometrically consistent structures. This constraint not only stabilizes training but also encourages the network to encode focus-aware, surface-consistent representations that align with real-world depth geometry.

**Importance of Focus-aware Weighting on $L_{sv}$.** We evaluate the contribution of the sharpness-based focus weighting $q$ in our spatial variational loss $L_{sv}$, which adjusts the influence of each pixel according to its sharpness. Removing $q$ or inverting its effect to prioritize defocused regions (i.e., using $1-q$) results in consistent performance drops. These ablations show that emphasizing reliable in-focus pixels is critical for guiding the model toward accurate depth learning. By attenuating supervision in out-of-focus regions, the focus-aware weighting encourages the network to concentrate on spatial regions that are both photometrically informative and structurally meaningful.

Additional ablation results, runtime analysis, and quantitative comparisons are provided in Appendices C and E.

## 5 Conclusion

We introduce DualFocus, a novel DFF framework that exploits the intrinsic characteristics of focal stack data by jointly modeling spatial and focal variations through a dual variational formulation. Our method explicitly incorporates focus-dependent gradient behaviors and focal sharpness distributions to distinguish true geometric structures from texture-induced ambiguities—an area where previous DFF approaches often fall short. By leveraging the complementary strengths of spatial and focal constraints, our framework delivers robust and accurate depth predictions, particularly in scenes with fine textures or sharp depth transitions. Extensive experiments across four benchmark datasets validate the effectiveness of our approach, setting a new standard in focus-based depth estimation.

**Limitation.** While DualFocus improves robustness under challenging focus conditions, limitations remain. First, in regions with low texture contrast, focus variation may offer weak depth cues. Second, the method assumes spatially aligned focal stacks, making it sensitive to motion-induced misalignment. Finally, the scene-dependent blur characteristics (e.g., lighting artifacts or lens-specific aberrations) may limit the model's generalization. Addressing these limitations through motion-invariant modeling or alignment-free representations is a promising direction for future work.

## 6   Acknowledgement

This research was supported by the National Research Foundation of Korea (NRF) grant funded by the Korea government (MSIT)(No. RS-2024-00340745) and the Yonsei Signature Research Cluster Program of 2025 (2025-22-0013).

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

# A Implementation Details

## A.1 Training Details

Our model is trained end-to-end using the Adam optimizer with cosine annealing for learning rate scheduling. We use a focal stack of $N = 5$ images as input, and adopt a ResNet-18 FPN [13] as the encoder and 3D-ResNet blocks [8] as the decoder. The encoder produces the focus volume $V^*$ with 64-channel features ($2C_1 = 64$), while the variational module operates on 16-channel features ($C_2 = 16$). Two variational loss terms are weighted by $\lambda_{\text{sv}} = 20$ and $\lambda_{\text{fv}} = 100$, balancing spatial constraints and focus probability constraints, respectively. For training, we use a batch size of 16 for NYU Depth V2 and 20 for other datasets, with training durations of 40 and 2000 epochs, respectively. A full list of model and training hyperparameters is summarized in Table 6.

Table 6: Model and training hyperparmeters.

| Hyperparameter | Value |
|---|---|
| Focal stack size $N$ | 5 |
| Encoder | Resnet-18 FPN [13] |
| Decoder | 3D-ResNet blocks [8] |
| Feature channel $C_1$ | 32 |
| Feature channel $C_2$ | 16 |
| $\lambda_{\text{sv}}$ | 20 |
| $\lambda_{\text{fv}}$ | 100 |
| Optimizer | Adam ($\beta 1 = 0.9, \beta 2 = 0.999$) |
| Scheduler | Cosine annealing |
| Initial learning rate | 0.001 |
| Batch size | 16 / 20 (NYU dataset / Others) |
| Training epochs | 40 / 2000 (NYU dataset / Others) |

## A.2 Focal Stack Data Synthesis

To address the limitations of datasets lacking real focal stacks and enable robust comparisons with state-of-the-art (SOTA) models, we adopted a data synthesis method of prior work [6] to artificially generate focal stacks from a single image with ground truth depth. The synthesis process involves configuring a virtual camera with adjustable focus settings, defining specific focus distances to simulate depth-based focusing, and applying a circular kernel to introduce blur based on the ground truth depth and focus distances. The extent of blur, namely the circle of confusion (CoC), is calculated using a widely adopted equation [15, 24], which is expressed as

$$c = \frac{|S_2 - S_1|}{S_2} \cdot \frac{f^2}{N \times (S_1 - f)},$$

(9)

where $f$ is the focal length, $N$ is the f-number, $S_1$ is the in-focus subject distance, and $S_2$ is the out-of-focus distance. Following the process in prior work *HybridDepth* [6], we slightly cropped the NYU Depth v2 images by removing borders from the depth maps to mitigate issues in focal stack creation. For detailed implementation, readers are referred to [6].

## A.3 Model Architecture

The proposed model is a convolutional encoder-decoder network for depth estimation from a stack of $N$ focal images. It consists of four sequential components: an encoder, a variational module, a decoder with fusion modules, and a focus probability prediction head.

**Encoder.** The encoder, based on a ResNet-18 Feature Pyramid Network (FPN), processes the input focal stack using 2D convolutions and extracts multi-scale feature maps at 1/4, 1/8, 1/16, and 1/32 of the input resolution $h \times w$. At each scale, the features are reorganized into a 4D focus volume, following the approach of DFV [29], to encode focus-dependent information across focal planes.

**Variational Module.** To impose spatial variational constraints, a variational module operates at the 1/32 scale and estimates 16-channel gradient fields $[\Gamma^x, \Gamma^y] \in \mathbb{R}^{2 \times h/32 \times w/32 \times 16 \times N}$ from the focus volume. These gradients are used to solve a least-squares problem, yielding surface fields $z^* \in \mathbb{R}^{h/32 \times w/32 \times 16 \times N}$. The surface features are regularized via group normalization, treating the 16 channels as a single group to encourage implicit surface consistency. A convolutional layer then expands the channel dimension from 16 to 128 for richer representation.

**Decoder with Fusion Modules.** The decoder comprises three hierarchical fusion modules that operate at progressively higher spatial resolutions: 1/16, 1/8, and 1/4 of the input size. At each scale, a fusion module integrates two complementary streams of information: the focus volume features extracted by the encoder and the upsampled surface features $z^*$ produced by the variational module. To effectively merge these modalities, the fusion is performed using 3D convolutions, which can jointly process spatial and focal dimensions. This design allows the network to refine depth-relevant features by leveraging both the focus-specific sharpness variations and the geometric surface cues estimated at coarser levels. As the decoder progresses through the scales, the features become increasingly detailed and structurally coherent. Finally, the refined outputs from all fusion stages are upsampled to a common resolution and concatenated, yielding a unified multi-scale representation that encapsulates rich depth cues across both spatial and focal dimensions. This multi-scale representation serves as the input to the subsequent focus probability prediction head.

**Focus Probability Prediction.** The aggregated multi-scale features are further upsampled to match the resolution of the input images and then passed through the focus prediction head, which outputs a probability map $p \in \mathbb{R}^{h \times w \times N}$. A softmax is applied across the $N$ focal planes at each pixel location. The final per-pixel depth is then calculated as the expectation over the known focal distances $\{f_n\}$, weighted by the predicted focus probabilities $p_n$: $\widehat{D}(\mathbf{x}) = \sum_{n=1}^{N} p_n(\mathbf{x}) f_n$. This procedure allows the network to produce continuous depth estimates from discrete focal planes, effectively integrating the focus information learned across the entire stack.

**Dual Variational Supervision.** During training, the model is supervised with two complementary variational losses. The spatial variational loss computes local gradients from the reconstructed surface features and compares them to the ground-truth depth gradients in in-focus regions, guiding the network to produce coherent surface structures. The focal variational loss constrains the predicted in-focus probabilities across the focal stack to follow a smooth and physically consistent transition. These two losses are applied alongside the standard depth regression loss, ensuring the network effectively learns both spatial and focus-dependent cues from the multi-focus input.

### A.4 Evaluation Metrics

We evaluate our method using a set of widely adopted depth estimation metrics that collectively measure the accuracy, robustness, and perceptual quality of predicted depth maps. Specifically, we report Mean Squared Error (MSE), Root Mean Squared Error (RMSE), Absolute Relative Error (AbsRel), Squared Relative Error (SqRel), accuracy at multiple thresholds ($\delta_1$, $\delta_2$, $\delta_3$), and Bumpiness (Bump). Let $D = \{d_i\}$ denote the predicted depth map and $D^* = \{d_i^*\}$ the corresponding ground-truth depth map, where $i$ indexes the $N$ valid pixels in the image (e.g., pixels where ground-truth depth is available and non-zero).

**MSE** measures average squared differences, emphasizing larger errors: $\frac{1}{N} \sum_{i=1}^{N} (d_i - d_i^*)^2$.

**RMSE** gives average error in depth units: $\sqrt{\frac{1}{N} \sum_{i=1}^{N} (d_i - d_i^*)^2}$.

**AbsRel** averages absolute differences over ground-truth depth: $\frac{1}{N} \sum_{i=1}^{N} \frac{|d_i - d_i^*|}{d_i^*}$.

**SqRel** averages squared differences over ground-truth depth: $\frac{1}{N} \sum_{i=1}^{N} \frac{(d_i - d_i^*)^2}{d_i^*}$.

**Accuracy ($\delta_k$)** is pixels with depth ratio $< 1.25^k$ ($k = 1, 2, 3$): $\frac{1}{N} \sum_i \mathbb{1}\big(\max(\frac{d_i}{d_i^*}, \frac{d_i^*}{d_i}) < 1.25^k\big)$.

**Bumpiness (Bump)** assesses smoothness via the variance of the discrete Laplacian: $\frac{1}{N} \sum_{i=1}^{N} (\nabla^2 d_i)^2$.

# B  Optimal Solution Derivation for Least-Squares System

Given the predicted spatial gradients $\Gamma_n^{(c)} \in \mathbb{R}^{2HW}$ and a first-order difference operator $P \in \{-1, 0, 1\}^{2HW \times HW}$, we aim to recover the surface field $z_n^{(c)} \in \mathbb{R}^{HW}$ that best satisfies the gradient constraint:

$$P z_n^{(c)} = \Gamma_n^{(c)}. \tag{10}$$

Since the system is overdetermined — i.e., the number of equations ($2HW$) exceeds the number of unknowns ($HW$) — we seek a least-squares solution by minimizing the residual:

$$z_n^{*(c)} = \arg\min_z \|P z_n^{(c)} - \Gamma_n^{(c)}\|_2^2. \tag{11}$$

This objective is convex, and its minimum can be obtained by setting the gradient with respect to $z$ to zero. The closed-form solution is then given by the Moore-Penrose pseudoinverse:

$$z_n^{*(c)} = (P^\top P)^{-1} P^\top \Gamma_n^{(c)}. \tag{12}$$

We refer readers to [14] for more details on the derivation.

# C  Additional Experimental Results

## C.1  Spatial Resolution of the Gradient Fields $\Gamma$ and Surface Fields $z^*$

To analyze the trade-off between performance and computational efficiency, we investigate the spatial resolution of the gradient fields $\Gamma$ and surface fields $z^*$. As shown in Table 7, increasing the resolution from $10 \times 10$ to $20 \times 20$ consistently improves accuracy, particularly in RMSE and SqRel. However, the performance gain between $14 \times 14$ and $20 \times 20$ is marginal, while the runtime increases from 26ms to 32ms. Our final model adopts a $14 \times 14$ resolution, which achieves near-optimal accuracy with lower computational cost, making it a more efficient and practical choice for real-world applications.

Table 7: Ablation study on the spatial resolution of the gradient fields $\Gamma$ and surface fields $z^*$ on the NYU Depth v2 dataset. Increasing resolution improves accuracy but increases runtime.

| Resolution | RMSE ↓ | log RMSE ↓ | AbsRel ↓ | SqRel ↓ | Runtime (ms) ↓ |
|---|---|---|---|---|---|
| $10 \times 10$ | 0.079 | 0.022 | 0.014 | 0.0024 | 25 |
| $14 \times 14$ | 0.075 | **0.020** | **0.013** | 0.0021 | 26 |
| $20 \times 20$ | **0.073** | **0.020** | **0.013** | **0.0020** | 32 |

## C.2  Feature Channel $C_2$ of the Gradient Fields $\Gamma$ and Surface Fields $z^*$

To investigate the impact of feature channel dimensionality in our geometric field representations, we conduct an ablation study on the number of feature channels $C_2$ used in both the gradient fields $\Gamma$ and surface fields $z^*$. As presented in Table 8, increasing the channel size from 1 to 16 leads to consistent improvements across all evaluation metrics on the DDFF 12-Scene dataset.

Specifically, the best performance is achieved when $C_2 = 16$, which suggests that a richer feature representation in the geometric fields enables the model to better capture fine-grained focus cues and surface details. Conversely, overly limited channel dimensions (e.g., $C_2 = 1$) may restrict the expressiveness of the learned gradient and surface structures, leading to suboptimal depth estimates.

Table 8: Ablation study on the number of feature channels $C_2$ used in the gradient fields $\Gamma$ and surface fields $z^*$ on the DDFF 12-Scene dataset.

| Feature Channel $C_2$ | RMSE ↓ | log RMSE ↓ | AbsRel ↓ | SqRel ↓ |
|---|---|---|---|---|
| 1 | 0.0214 | 0.226 | 0.188 | 0.0067 |
| 8 | 0.0203 | 0.216 | 0.174 | 0.0060 |
| 16 | **0.0195** | **0.200** | **0.162** | **0.0057** |

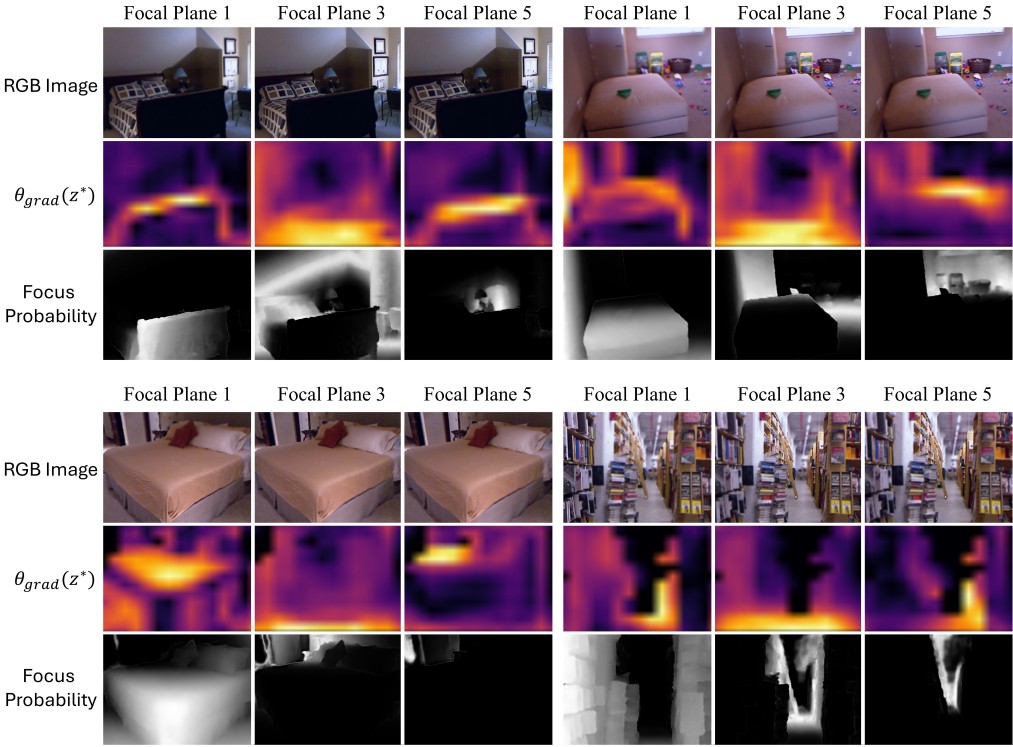

Figure 5: Visualization of input image, surface gradient magnitude $\|\theta_{\mathrm{grad}}(z_n^*)\|$, and predicted focus probability $p$ at focal planes 1, 3, and 5 (out of $N = 5$).

## C.3 Visualization of Implicit Surface Field Gradients

The proposed method leverages an implicit surface field $z^* \in \mathbb{R}^{H \times W \times C_2 \times N}$ to encode multi-channel surface representations across the focal stack. As described in Section 3.3 of the main paper, $z^*$ is derived by projecting predicted gradient features onto the space of integrable surfaces, effectively reconstructing plausible local geometry per focal plane. Since $z^*$ resides in a multi-dimensional latent space, directly interpreting its content is non-trivial. To provide insights into its internal structure and focus dependence, we visualize the spatial gradients $\theta_{\mathrm{grad}}(z_n^*)$, which are used in the spatial variational loss $L_{\mathrm{sv}}$.

Figure 5 presents representative examples from three focal planes (indices 1, 3, and 5, where $N = 5$) and includes: (1) the RGB image at each focal plane, (2) the magnitude of $\theta_{\mathrm{grad}}(z_n^*)$, and (3) the predicted focus probability $p$. The visualizations reveal a strong alignment between focus probability and gradient coherence that in-focus regions with high $p_n$ values consistently exhibit sharper and more structured gradient patterns, whereas defocused regions result in weaker or noisy gradients. This focus-aware behavior emerges despite no explicit supervision on sharpness or texture—indicating that the integrable reconstruction process implicitly guides the model to concentrate gradient energy in geometrically reliable regions.

This empirical evidence supports the underlying design motivation introduced in the main paper: namely, that predicting spatial gradients rather than absolute depths allows the network to exploit focus-based depth variations and better distinguish true geometric discontinuities from texture-induced noise. The consistency and structure of the gradients in sharp regions demonstrate that the model learns to encode geometry in a focus-aware manner. Furthermore, the observed variation in gradient responses across focal planes shows that the network effectively leverages multi-plane comparison to isolate reliable depth cues.

In summary, this analysis reinforces the role of $z^*$ not just as an intermediate representation, but as a focus-conditioned surface descriptor. The structured, integrable gradients emerging in high-focus regions validate our spatial variational formulation and highlight the importance of incorporating spatial constraints tailored to the unique characteristics of DFF tasks.

Table 9: Impact of focal variational constraints on focus probability distribution and performance.

| Dataset | $L_{fv}$ | Invalid Focus Trend (%) ↓ | RMSE ↓ | log RMSE ↓ | AbsRel ↓ |
|---|---|---|---|---|---|
| NYU Depth v2 | ✗ | 46.2 | 0.078 | 0.022 | 0.014 |
| NYU Depth v2 | ✓ | **6.2** | **0.075** | **0.020** | **0.013** |
| DDFF 12-Scene | ✗ | 88.2 | 0.021 | 0.211 | 0.167 |
| DDFF 12-Scene | ✓ | **13.4** | **0.020** | **0.200** | **0.162** |

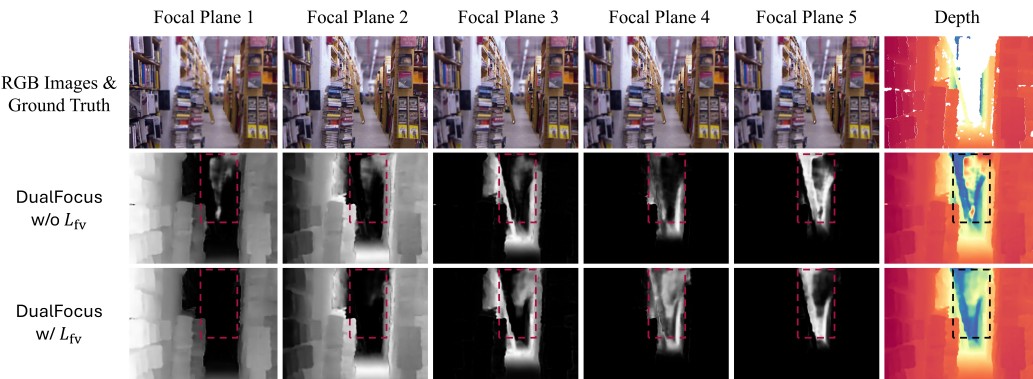

Figure 6: Impact of the focal variational constraint $L_{fv}$ on focus probability and depth prediction. Without $L_{fv}$, the model produces inconsistent focus probabilities across focal planes (e.g., non-monotonic trend in red-boxed region), leading to depth errors. With $L_{fv}$, the focus trend is regularized to be monotonic, resulting in more accurate and stable depth maps.

## C.4 Effect of the Focal Variational Constraints

Table 9 presents the quantitative effect of applying the focal variational loss $L_{fv}$ on both the NYU Depth v2 and DDFF 12-Scene datasets. The metric "Invalid Focus Trend" measures the percentage of pixels whose predicted focus probability does not exhibit a monotonic decrease with respect to the peak focal plane, i.e., cases where the focus probability fails to drop off smoothly as the distance from the most in-focus slice increases. Lower values indicate better focus behavior that aligns with the expected defocus pattern in natural focal stacks.

Without the proposed variational constraint, a substantial portion of pixels—46.2% in NYU and 88.2% in DDFF—exhibit invalid focus trends. By incorporating $L_{fv}$, these rates drop significantly to 6.2% and 13.4%, respectively, demonstrating that the constraint effectively enforces physically consistent focus probability distributions. This improvement in focus consistency also translates into slight gains in depth accuracy across all evaluated metrics (RMSE, log RMSE, and AbsRel), confirming that encouraging structured focus behavior leads to more reliable depth estimation.

To further illustrate the effect of the focal variational constraint, we visualize the predicted focus probability distributions across the focal stack ($N = 5$) along with the corresponding depth maps. As shown in Figure 6, DualFocus without $L_{fv}$ exhibits an invalid focus trend in the red-boxed region. Specifically, the focus probability peaks at focal plane 5 and decreases across planes 2 to 4, as expected. However, focal plane 1—being furthest from the peak—unexpectedly shows a higher probability than the intermediate planes. This violates the expected monotonic decay pattern and leads to an incorrect depth prediction in the affected region. In contrast, DualFocus with $L_{fv}$ produces a smooth and monotonic focus probability curve centered around the most in-focus plane. The probability consistently decreases as the focal plane moves away from the peak, resulting in a valid focus trend. This improved structure enables the model to infer a more accurate and geometrically consistent depth map.

These qualitative results support the quantitative findings in Table 9, demonstrating that the focal variational constraint not only enforces physical plausibility in the focus distribution but also leads to more reliable depth estimation in practice. We can observe that this approach reduces errors caused by invalid focal trends across diverse scenarios, with additional examples provided in Figure 7.

## C.5 Scalability to Focal Stack Size *N*

In this section, we evaluate the scalability of our method with respect to the number of focal stack images on the NYU Depth v2 dataset. Table 10 summarizes the performance of our method compared to prior DFF approaches, including HybridDepth [6] and DFV [29], as well as a recent single-image depth estimation method, Depth Anything [30].

The number of focal planes in DFF tasks directly affects the richness of depth cues available for learning. As expected, performance generally improves with more inputs and degrades with fewer, a trend observed in both DFV and our method.

Despite using fewer focal planes, our method demonstrates strong scalability. With only 3 focal images, it outperforms DFV and surpasses HybridDepth trained with 5 images, as well as the single-image Depth Anything model trained on a large corpus of general-purpose depth data. This result highlights the effectiveness of our approach in extracting meaningful depth cues even under sparse focus supervision.

When increasing the number of focal planes to 10, our method achieves a substantial performance gain, establishing a new state of the art across all metrics. This demonstrates that our model can effectively leverage sparse focal stacks while fully capitalizing on rich focus information when available, resulting in high-quality and geometrically consistent depth predictions.

Table 10: Performance comparison on NYU Depth v2 with varying focal stack size *N*. Hybrid-Depth [6] results under the 3-stack setting are not reported due to reproducibility issues in the released code.

| Method | Focal Stack Size *N* | RMSE ↓ | AbsRel ↓ |
|---|---|---|---|
| Depth Anything [30] | 1 | 0.206 | 0.056 |
| HybridDepth [6] | 3 | - | - |
| DFV [29] | 3 | 0.134 | 0.030 |
| Ours | 3 | 0.119 | 0.023 |
| HybridDepth [6] | 5 | 0.128 | 0.026 |
| DFV [29] | 5 | 0.094 | 0.020 |
| Ours | 5 | 0.075 | 0.013 |
| HybridDepth [6] | 10 | 0.083 | 0.015 |
| DFV [29] | 10 | 0.069 | 0.011 |
| Ours | 10 | 0.052 | 0.008 |

## C.6 Model Size and Runtime

To assess the efficiency of our method, we compare the model size and inference time against both single-image depth estimation (SIDE) and depth-from-focus (DFF) baselines, as summarized in Table 11. Our model achieves a balanced design with a small parameter count and fast runtime. Compared to other DFF-based methods, it remains lightweight and efficient, while delivering superior accuracy. These results demonstrate the practicality of our method for real-time depth estimation using focal stacks.

Table 11: Comparison of model size and runtime on the DDFF 12-Scene dataset. All measurements were conducted on a single NVIDIA RTX A6000 GPU.

| Method | Type | Focal Stack Size *N* | #Params (M) | Runtime (ms) |
|---|---|---|---|---|
| ZoeDepth-M12-N [3] | SIDE | 1 | 335 | 189 |
| Depth Anything (ViT-L) [30] | SIDE | 1 | 336 | 113 |
| Depth Anything (ViT-S) [30] | SIDE | 1 | 25 | 37 |
| DFV [29] | DFF | 5 | 20 | 16 |
| HybridDepth [6] | DFF | 5 | 66 | 41 |
| Ours | DFF | 5 | 27 | 28 |

# D   Discussion on Asymmetric Defocus Blur

In practical imaging systems, defocus blur can exhibit mild asymmetry due to lens imperfections and sensor characteristics. The proposed depth-from-focus framework is designed to remain robust under such conditions. The spatial variational constraint captures gradient variations induced by focus changes without assuming any parametric blur model, allowing the network to extract depth-relevant structural cues directly from the observed data. Complementarily, the focal variational constraint encourages a smooth unimodal distribution of predicted focus probabilities across the focal stack, reflecting the physical principle that sharpness peaks at the in-focus plane and decreases for neighboring planes. This behavior is preserved even when blur is moderately asymmetric, ensuring that the model can reliably identify in-focus regions.

Empirical evaluation across datasets with varying blur characteristics supports this robustness. Synthetic datasets such as NYU Depth v2 [25] and ARKitScenes [2] employ idealized symmetric defocus kernels, while FOD500 [15] also assumes circular blur. In contrast, DDFF 12-Scene [9] provides real-world focal stacks captured with a light-field camera, naturally introducing mild asymmetry. The framework maintains strong performance across all these datasets, including DDFF 12-Scene, demonstrating effective handling of moderate asymmetry.

However, extreme asymmetry or highly non-Gaussian point spread functions, which are uncommon in the considered datasets, may challenge generalization. Future work could explore explicit modeling of such scenarios to further enhance robustness.

# E   More Qualitative Comparisons

To further demonstrate the effectiveness of our method, we present additional qualitative comparisons on three publicly available datasets: NYU Depth v2 [25], DDFF 12-Scene [9], and FoD500 [15]. As illustrated in Figures 8–10, our approach consistently produces more coherent depth maps compared to previous state-of-the-art methods, HybridDepth [6] and DFV [29].

All visualizations include data from publicly available datasets. Human subjects, where visible, are rendered in a non-identifiable form with facial features removed.

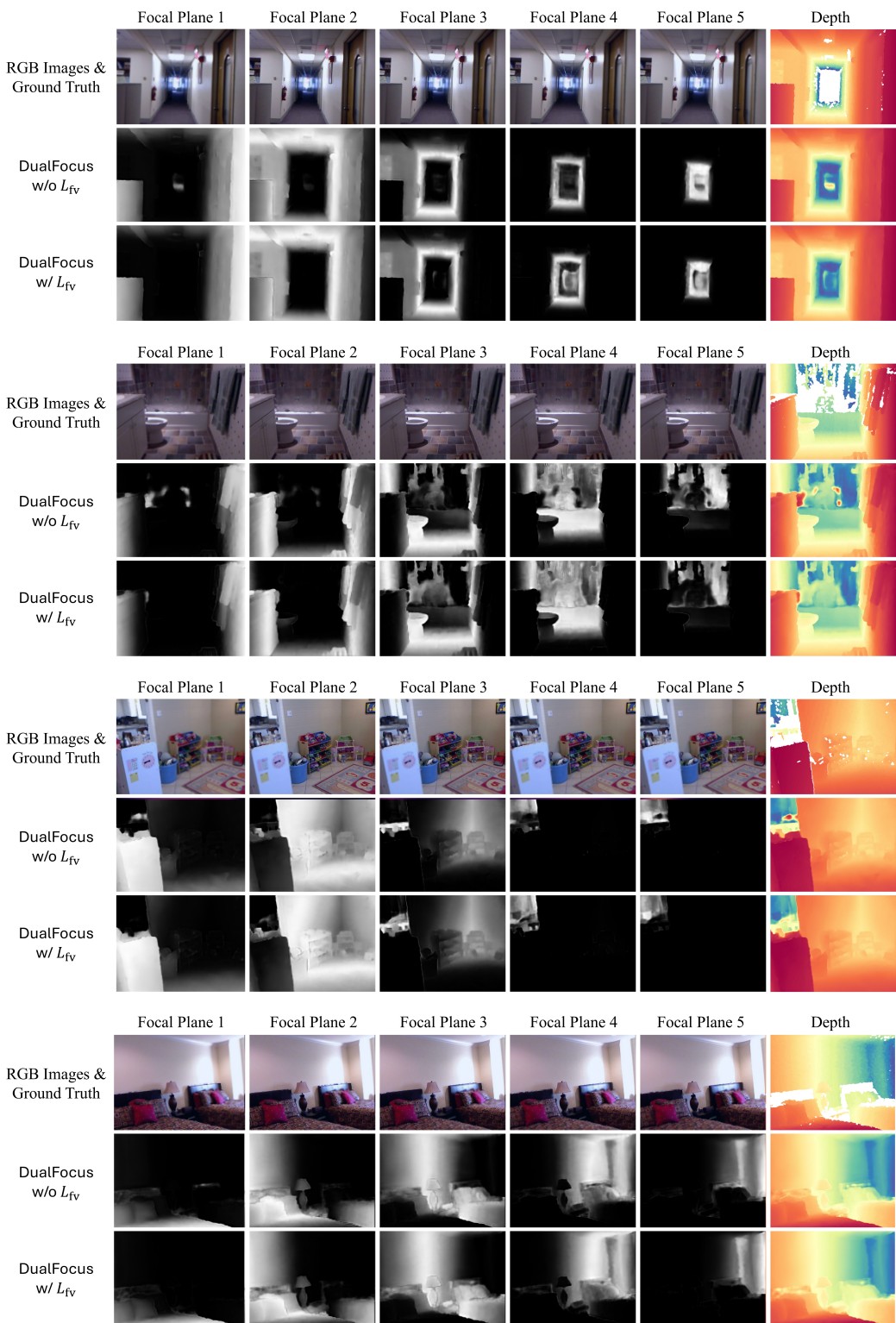

Figure 7: Effect of the focal variational constraints on the NYU Depth v2 dataset.

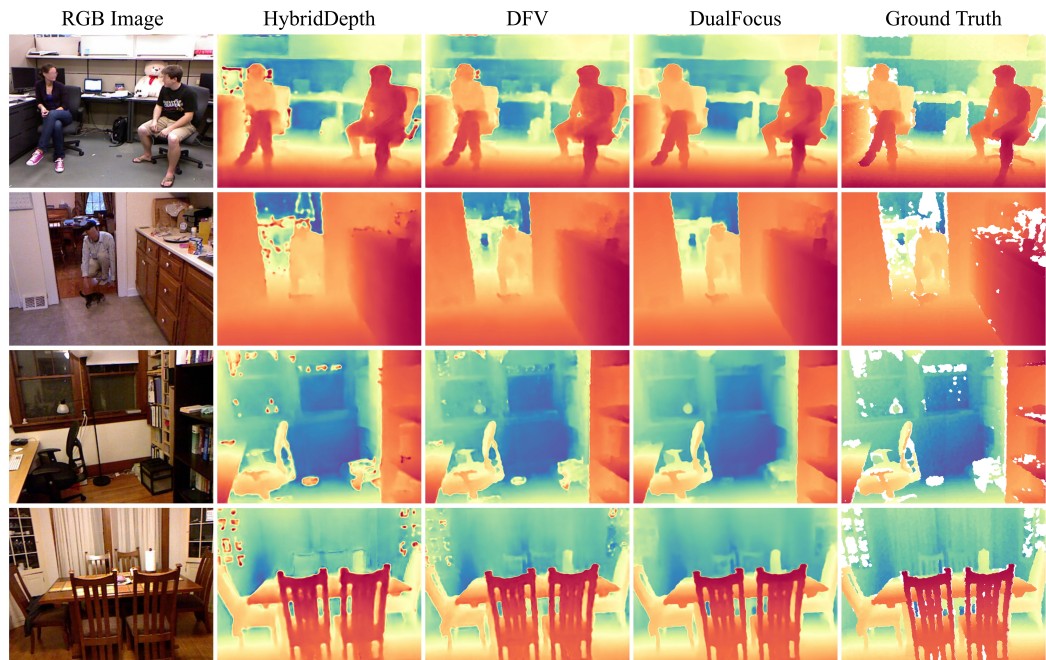

Figure 8: Qualitative comparison on the NYU Depth v2 dataset.

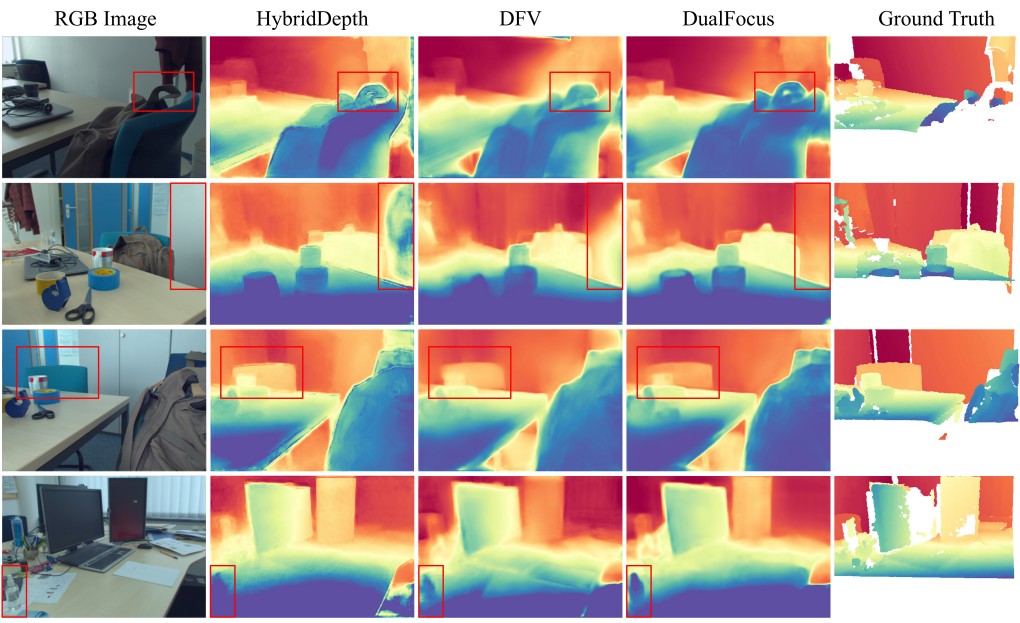

Figure 9: Qualitative comparison on the DDFF 12-Scene dataset.

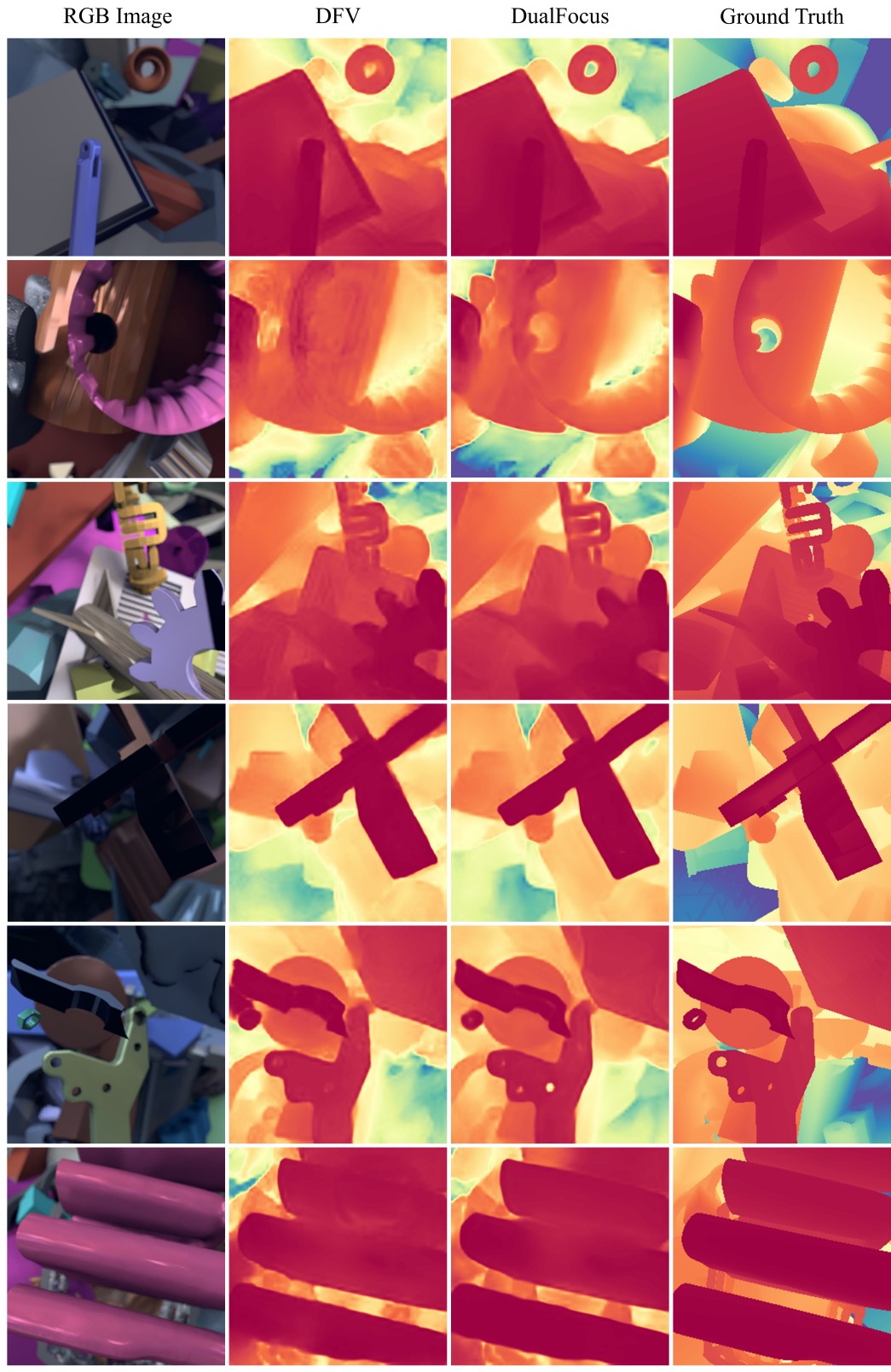

Figure 10: Qualitative comparison on the FoD500 dataset.

