# OpenReview forum: "DualFocus: Depth from Focus with Spatio-Focal Dual Variational Constraints"
_NeurIPS.cc/2025/Conference — NeurIPS 2025 poster_

### Official Review · Reviewer_aqEp · 2025-06-29

**Clarity:** 3
**Significance:** 3
**Originality:** 3
**Rating:** 5
**Confidence:** 4

**Summary:**

This paper presents DualFocus, a novel Depth-from-Focus (DFF) framework that addresses the limitations of prior methods in complex scenes with fine textures or abrupt depth changes. The approach introduces spatial and focal variational constraints to model gradient patterns across focal stacks, leveraging physical focus behavior to distinguish true depth edges from texture artifacts. Experiments on four datasets (NYU Depth v2, FoD500, DDFF 12-Scene, ARKitScenes) demonstrate state-of-the-art depth accuracy and generalization, outperforming prior DFF methods. The key contributions include the dual-constraint formulation, a unified differentiable architecture, and robust performance across diverse scenarios.

**Questions:**

1.	Have you tested the impact of the number of images in the focal stack on the performance?

**Ethical Concerns:**

["NO or VERY MINOR ethics concerns only"]

**Final Justification:**

I have upgraded the rating.

**Limitations:**

Same with weakness

**Quality:**

3

**Strengths And Weaknesses:**

Strengths*
1.Innovative Dual Constraints: The spatial and focal variational constraints effectively integrate optical principles, improving robustness in challenging regions by enforcing gradient integrability and unimodal focus distributions .
2. Intuitive Motivation：Using the gap between focal distance and true depth to model clarity, employing this as a weight to supervise gradients, and generating probabilities based on clarity to weight and compute the final depth aligns with human intuition.
3. Comprehensive Experimental Validation: Thorough evaluations on synthetic and real-world datasets, including zero-shot transfer to ARKitScenes, validate generalizability.

Weaknesses*
1. The accuracy is highly dependent on the number of focal stacks. If the number of focal stacks is small, it may lead to inaccurate depth estimation.
2. The estimation of probability heavily relies on gradient-related surface features, which may limit its effectiveness in low-texture regions.
3. It seems that experiments were only conducted in indoor scenes without considering outdoor scenarios. And the proposed method in the paper has been fully validated through experiments.

---

> ### Author Rebuttal · Authors · 2025-07-28
>
> We thank the reviewer for the constructive suggestions and positive reception of our work. We list our response to your questions below.
> - - -
> ## 1. Scalability to Focal Stack Size
> We thank the reviewer for this insightful question regarding the scalability of our model with respect to the number of focal stack images. To address this, we conducted additional experiments on the NYU Depth v2 dataset using 3 and 10 focal planes. The table below summarizes the performance of our method compared to prior DFF approaches, including DFV and HybridDepth, as well as a recent single-image depth estimation method, Depth Anything. HybridDepth results under the 3-stack setting are not reported due to reproducibility issues in the official training code.
>
> | Method | Focal Stack Size | RMSE&#8595; | AbsRel&#8595; |
> |-------------|:----------------:|:----:|:------:|
> | Depth Anything | 1  | 0.206 | 0.056 |
> | HybridDepth | 3 | - | - |
> | HybridDepth | 5 | 0.128 | 0.026 |
> | HybridDepth | 10 | 0.083 | 0.015 |
> | DFV | 3 | 0.134 | 0.030 |
> | DFV | 5 | 0.094 | 0.020 |
> | DFV | 10 | 0.069 | 0.011 |
> | Ours | 3 | 0.119 | 0.023 |
> | Ours | 5 | 0.075 | 0.013 |
> | Ours | 10 | 0.052 | 0.008 |
>
> In DFF tasks, the number of focal planes directly affects the richness of depth cues available for learning. As expected, performance increases with more inputs and degrades with fewer, a common trend that is clearly reflected in both DFV and our method.
>
> Nonetheless, our method demonstrates strong scalability. Even with only 3 focal images, it outperforms DFV and surpasses HybridDepth trained with 5 images, as well as the single-image method Depth Anything trained on a large corpus of general-purpose depth data. This highlights the effectiveness of our design in extracting meaningful depth cues even under sparse focus supervision.
>
> Notably, when increasing the number of focal planes to 10, our method achieves a substantial performance gain, establishing a new state of the art across all metrics. This demonstrates not only that our model can effectively utilize sparse focal stacks, but also that it fully capitalizes on rich focus information when available, leading to high-quality and geometrically consistent depth predictions.
>
> We appreciate the reviewer’s suggestion once again and are happy to include these additional experiments and clarifications in the final version of the paper.
> - - -
> ## 2. Concern regarding Performance in Low-Texture Regions
> We understand the reviewer’s concern that our method, which estimates focus probability based on gradient-related surface features, may be vulnerable in low-texture regions. As also described in Line 324 of the main paper, performance can indeed degrade when large spatial areas contain little to no texture, resulting in weak or absent focus-induced gradient variations. Importantly, such regions are known to be inherently challenging for depth estimation in general, including both single-image and prior DFF-based methods, due to the scarcity of discriminative visual cues.
>
> That said, we observe that our model still performs reasonably well in many low-texture scenarios. We attribute this to the design of our gradient-based constraints, which can respond to subtle focus transitions when present, and to the use of gradient-related features computed at a relatively lower spatial resolution. This lower-resolution representation helps mitigate the reliance on fine-grained local gradients, making the model more robust in textureless regions by allowing it to extract meaningful focus cues from broader structural differences across focal planes.
>
> While we are unable to include new figures under the NeurIPS rebuttal policy this year, several examples in our original submission demonstrate promising results in low-texture settings:
>
> - In the fourth row of Figure 4 in the main paper, the white recessed sink has smooth surfaces with very limited texture, particularly around its edge. Competing methods struggle to capture the depth discontinuity at the sink boundary, whereas our model produces a clearer and more structurally accurate prediction.
>
> - In Figure 3 of the supplementary material, two of the four scenes (bottom two rows) exhibit large textureless wall regions. Despite minimal texture, our model successfully identifies the in-focus areas across different focal planes and predicts depth values that closely match the ground truth.
>
> - In the second row of Figure 5 in the supplementary material, the white wall on the right side is textureless. While other prior methods exhibit significant noise in this region, our model remains robust and produces clean, stable depth predictions.
>
> We hope that these examples and clarifications help demonstrate the reasonable robustness of our method in low-texture regions, and that the overall modeling strategy helps mitigate some of the limitations typically associated with gradient-based approaches.
>
> - - -
> ## 3. Indoor-Only Evaluation
>
>
> We agree that exploring depth-from-focus in outdoor scenarios would be a valuable extension of current research. In practice, however, most existing DFF datasets and studies have been limited to indoor environments. This is largely due to the practical challenges of capturing focal stacks with consistent focus sweeps outdoors—such as variable lighting, motion artifacts, and the difficulty of obtaining accurate ground truth depth in large-scale, unstructured scenes. Consequently, our work, like most prior DFF approaches, focuses on indoor datasets where data collection and evaluation are more tractable.
>
> Nonetheless, we appreciate the reviewer’s comment, which highlights an important consideration for real-world applicability. We believe that expanding DFF research to more diverse and unconstrained environments, including outdoor scenes, represents a meaningful future direction.

---

> > ### Comment · Reviewer_aqEp · 2025-08-05
> > **Response to the author rebuttal**
> >
> > Thank you for your comprehensive and in-depth response to my questions. Regarding the scalability of focal stack size, the supplementary experiments clearly demonstrate the advantages of your method, especially its outstanding performance when the number of focal planes increases to 10, which strongly supports the viewpoints of the paper. For the concern about performance in low-texture regions, you have provided a reasonable explanation from the perspective of method design and illustrated it with existing examples, all of which reflect the certain advantages of the method. Regarding the issue of indoor-only evaluation, you have also given an objective analysis of the reasons and recognized the significance of extending the research to outdoor scenes. Overall, your responses have enhanced the persuasiveness and completeness of the paper. I acknowledge these supplementary contents and agree to include them in the final version of the paper.

---

> ### Author Response · Authors · 2025-08-05
>
> We're pleased that our response resolved your concerns, and we're grateful for your support in incorporating the supplementary contents into the final version.
> Thank you again for your time and insightful comments.

---

### Official Review · Reviewer_hJeu · 2025-06-30

**Clarity:** 2
**Significance:** 2
**Originality:** 3
**Rating:** 4
**Confidence:** 3

**Summary:**

This paper introduces DualFocus to estimate scene depth from focal stacks. Different from existing methods, DualFocus  leverages the unique gradient patterns across a focal stack by modeling both spatial and focal variations through dual variational constraints. The spatial constraint captures consistent gradient changes across focus levels to identify true depth edges, while the focal constraint ensures the predicted focus probabilities follow a physically plausible unimodal and monotonic distribution. Experiments show that the proposed constraints can help separate real geometry from texture artifacts, improving depth accuracy and robustness.

**Questions:**

- Can the authors demonstrate the model’s generalization ability by training on one dataset and testing on another? In the comparison, please add the results of single-image-based depth estimation methods like DepthAnything.
- The improvement over prior DFF methods (e.g., DFV on DDFF) is minimal. Can the authors provide ablation studies or visualizations that clarify where and why the gains occur?
- How does the method behave when using fewer or more than 5 focal images? Is the model robust to missing frames or variable stack sizes?

**Ethical Concerns:**

["NO or VERY MINOR ethics concerns only"]

**Final Justification:**

This paper introduce DualFocus, which leverages the unique gradient patterns across a focal stack by modeling both spatial and focal variations. My original rating is negative considering the tradeoff between depth estimation accuracy and the number of focal stacks required, and the marginal gain over existing methods on DDFF-12 dataset. The authors provide a comprehensive rebuttal and address my concerns. I suggest the authors put the corresponding discussion in the revised version.

**Limitations:**

Yes

**Quality:**

2

**Strengths And Weaknesses:**

Strength
- The DualFocus framework uses a physics-aware modeling approach that analyzes gradient structures and probability distributions across spatial and focal dimensions, and leverages the complementary strengths of spatial and focal constraints. The framework produces robust and accurate depth predictions, particularly in scenes with fine textures or sharp depth transitions. The paper is technically sound.
- The paper is easy to follow and well organized.


Weakness
- Each dataset is trained and tested independently, and there is no evidence of cross-dataset generalization. This raises concerns about the scalability of the method to real-world settings, especially compared to zero-shot monocular models, such as DepthAnything, VGGT.
- Unlike single-image methods such as DepthAnything, the proposed approach requires multiple focal stack images. Yet, its depth predictions—particularly in textured regions (see Fig. 6 in supp.)—remain suboptimal. Furthermore, it lacks qualitative comparisons with single-image baselines, and only shows marginal gains over DFV on the real-world DDFF dataset (Fig. 5 in supp.).
- The method operates with a fixed input of 5 focal images, but it remains unclear how robust or flexible it is when using more or fewer images. A study on scalability across focal stack sizes would strengthen the evaluation.

---

> ### Author Rebuttal · Authors · 2025-07-28
>
> We thank the reviewer for the detailed comments and valuable feedback. We list our response to your questions below.
> - - -
> ## 1. Cross-Dataset Generalization and Comparison with Zero-Shot SIDE Models
> We would like to clarify that the generalization ability of our model has already been evaluated in Section 4.3 of the main paper through zero-shot transfer experiments (please see Table 4). In this evaluation, all Depth-from-Focus (DFF) models, including ours, are trained exclusively on the NYU Depth v2 dataset and tested zero-shot on the ARKitScenes dataset without any additional training.
>
> This experiment also includes comparisons with four different zero-shot single-image depth estimation (SIDE) models—Depth Anything, ZoeDepth, DistDepth, and ZeroDepth. These models have been trained on large-scale and diverse datasets. Depth Anything is trained on 63.5 million images from 14 datasets, and ZoeDepth-M12-N is pretrained on 12 datasets and finetuned on NYU Depth v2. DistDepth is trained on 500K images collected from approximately 1,000 diverse environments, while ZeroDepth is trained on 30.1 million images from 5 datasets.
>
> Despite their extensive training, our method demonstrates competitive zero-shot performance on ARKitScenes. Notably, it outperforms all other DFF baselines—including HybridDepth, which leverages Depth Anything predictions as input—and achieves better RMSE than all SIDE baselines.
>
> These results underscore the robust generalization capability of our approach without relying on large-scale or diverse training data. Although we cannot provide qualitative comparisons here due to this year’s NeurIPS rebuttal policy, internal observations indicate our model tends to produce sharper object boundaries and more consistent depth across focus transitions on both NYU Depth v2 and ARKitScenes samples. While large-scale pretrained SIDE models such as Depth Anything may yield smoother and more visually coherent depth maps due to massive and diverse datasets, our method instead preserves physically grounded focus cues and local depth variations, which often lead to more faithful representations of true scene geometry, particularly near defocus transitions and fine structures.
>
> In response to the reviewer’s feedback, we will include qualitative comparisons with Depth Anything  model in the final version. We hope this sufficiently addresses the reviewer’s concern and demonstrates our model’s scalability and applicability to unseen domains.
> - - -
> ## 2. Marginal Gains over Prior DFF Methods
> We appreciate the reviewer’s concern and would like to offer additional context to better clarify the perceived gains over prior DFF methods, particularly on the DDFF-12 dataset (Fig. 5 in the supplementary material).
>
> While the overall visual quality on the DDFF-12 dataset may not appear strong, we believe this is largely attributable to the limited size and diversity of the dataset—only 600 focal stacks are available for training. This scarcity of data makes it challenging for all DFF models, including ours, to produce consistently detailed or visually compelling depth maps. In contrast, our primary benchmark, NYU Depth v2, contains 24,000 focal stacks, allowing for more stable training, better generalization, and thus clearer distinctions in model performance.
>
> Despite this limitation, our method still demonstrates consistent and clear quantitative improvements, as reported in Table 3 of the main paper. Compared to DFV, our approach yields an approximately 17.5% reduction in MSE (from $5.7 \times 10^{-4}$ to $4.7 \times 10^{-4}$), indicating a notable performance gain. Furthermore, we would like to draw attention to specific regions where our method exhibits meaningful qualitative gains that may not be immediately apparent without closer inspection.
>
> In Figure 5 of the supplementary material, we highlight several such cases:
>
> - **1st row:** The small loop on top of the bag has a hollow center. Our method captures its edge more accurately than prior methods.
>
> - **2nd row:** The textureless white wall on the right often leads to noise in prior methods, whereas our model produces a clean and stable depth map.
>
> - **3rd row:** The chair in the background exhibits a sharper depth edge in our result.
>
> - **4th row:** The transparent liquid container in the bottom left is clearly outlined by our method, while largely omitted by others.
>
> To better showcase these qualitative gains, we will revise the figure in the final version by adding zoomed-in views and annotations that guide attention to regions where differences are most evident. Additionally, we plan to include new examples where the improvements are more apparent, particularly in texture-sensitive areas and around depth discontinuities, thereby enhancing visual support for the reported gains.
>
> While the DDFF-12 dataset provides a challenging testbed due to limited diversity, we wish to emphasize that the broader evaluation results provide further evidence of our model's effectiveness. As shown in Tables 1–4 of the main paper, our method achieves consistent and noticeable improvements across all four datasets, surpassing state-of-the-art DFF approaches in diverse scenarios.
>
> Qualitative results further support these findings. For instance, Figure 4 of the main paper and Figures 4 and 6 of the supplementary materials demonstrate clear visual advantages on NYU Depth v2 and FoD500. Specifically:
>
> - In Figure 4 of the main paper, the first two rows show a carpet and a brown cabinet with repetitive circular patterns. These texture patterns confuse existing DFF methods, leading to erroneous depth in flat geometry regions. In contrast, our method produces consistent and smooth depth in this scene.
>
> - The last two rows show enhanced edge preservation in our results, highlighted by blue dotted boxes, which prior methods fail to retain.
>
> - In the fourth row of Figure 4 in the supplementary material, the window composed of small square glass panes contains complex reflections and visual clutter. Prior methods produce inconsistent errors in these regions, but our method regularizes such noise more effectively through the proposed focus-aware gradient-based constraints.
>
> - In the second row of Figure 6 in the supplementary material,  a vertically textured wooden cylinder with strong lighting variations is present. The previous work DFV predicts noisy depth in this area, while ours yields more consistent and clean predictions.
>
> To clarify where and why the gains occur, we kindly refer the reviewer to the comprehensive ablation studies in Table 5 of the main paper, which demonstrate the effectiveness of each proposed component. For instance, removing the spatial variational constraint alone results in a clear degradation in all metrics, underscoring its role in capturing focus-dependent spatial gradients and their variation along focal planes. Similarly, we analyze the effect of the focal variational constraint in Section D.4 of the supplementary material (Table 4 and Figures 2–3), showing that its absence leads to inconsistent depth transitions across focus levels and increased noise in textureless regions. These results confirm that enforcing a physically plausible focus distribution leads to sharply localized depth discontinuities and the suppression of artifacts in defocus-dominated areas. Together, these studies demonstrate that our model’s architectural inductive biases promote localized depth transitions and effectively mitigate texture-induced noise, leading to the consistent gains observed across benchmarks.
>
> We hope that the provided analysis, along with the planned clarifications in the final version, sufficiently address the reviewer’s inquiry regarding the gains.
> - - -
> ## 3. Scalability to Focal Stack Size
> We thank the reviewer for this insightful question regarding the scalability of our model with respect to the number of focal stack images. To address this, we conducted additional experiments on the NYU Depth v2 dataset using 3 and 10 focal planes. The table below summarizes the performance of our method compared to prior DFF approaches, including DFV and HybridDepth, as well as a recent single-image depth estimation method, Depth Anything. HybridDepth results under the 3-stack setting are not reported due to reproducibility issues in the official training code.
>
> | Method | Focal Stack Size | RMSE&#8595; | AbsRel&#8595; |
> |-|:-:|:-:|:-:|
> | Depth Anything | 1 | 0.206 | 0.056 |
> | HybridDepth | 3 | - | - |
> | HybridDepth | 5 | 0.128 | 0.026 |
> | HybridDepth | 10 | 0.083 | 0.015 |
> | DFV | 3 | 0.134 | 0.030 |
> | DFV | 5 | 0.094 | 0.020 |
> | DFV | 10 | 0.069 | 0.011 |
> | Ours | 3 | 0.119 | 0.023 |
> | Ours | 5 | 0.075 | 0.013 |
> | Ours | 10 | 0.052 | 0.008 |
>
> In DFF tasks, the number of focal planes directly affects the richness of depth cues available for learning. As expected, performance increases with more inputs and degrades with fewer, a common trend that is clearly reflected in both DFV and our method.
>
> Nonetheless, our method demonstrates strong scalability. Even with only 3 focal images, it outperforms DFV and surpasses HybridDepth trained with 5 images, as well as the single-image method Depth Anything trained on a large corpus of general-purpose depth data. This highlights the effectiveness of our design in extracting meaningful depth cues even under sparse focus supervision.
>
> Notably, when increasing the number of focal planes to 10, our method achieves a substantial performance gain, establishing a new state of the art across all metrics. This demonstrates not only that our model can effectively utilize sparse focal stacks, but also that it fully capitalizes on rich focus information when available, leading to high-quality and geometrically consistent depth predictions.
>
> We appreciate the reviewer’s suggestion once again and are happy to include these additional experiments and clarifications in the final version of the paper.

---

> > ### Comment · Reviewer_hJeu · 2025-08-01
> >
> > I appreciate the authors for their detailed rebuttal, which has addressed my concerns.
> > I would be interested in seeing the results using only two focal planes. This comparison would help clarify the effectiveness of the additional information provided by focal stacks relative to the single-image setting used in DepthAnything.

---

> > > ### Comment · Area_Chair_exW6 · 2025-08-01
> > >
> > > I consulted the PCs, and they say that "Discussions are meant to clarify understanding of existing materials, not asking for more new materials." In light of this, Reviewer hJeu: are there other questions that you would like to pose for more clarification other than asking for new results? Authors: would you be able to respond to Reviewer hJeu's inquiry at some level without having to generate additional results?
> > >
> > > Thank you.
> > >
> > > AC for Submission5350

---

> > > > ### Author Response · Authors · 2025-08-02
> > > >
> > > > We are pleased to hear that our previous response addressed the reviewer’s concerns. We appreciate Reviewer hJeu’s interest in understanding the relative effectiveness of focal stacks over single-image methods. While we understand that the current discussion period does not permit the introduction of new experimental results, we would like to note that we are currently exploring the suggested two-plane setting and intend to include these findings in our final supplementary material for greater transparency and completeness.
> > > >
> > > > We believe that our spatial variational constraint can still provide meaningful supervision in the two-plane case, particularly when the focal distances are sufficiently separated to reflect the scene’s geometry. Although the constraint was originally designed to leverage richer variation across multiple focal images, the focus-induced differences between in-focus and out-of-focus regions remain distinguishable even with two well-spaced planes. In-focus regions tend to exhibit consistent and integrable gradient patterns, forming coherent surfaces, whereas out-of-focus regions produce noisy and non-integrable gradients, leading to irregular surfaces. These contrasting characteristics can still provide valuable guidance to the model during training. While some performance degradation is expected due to the reduced diversity of focus cues, we anticipate that the model will outperform single-image methods, which must infer depth without any direct focus information.
> > > >
> > > > On the other hand, our focal variational constraint is less applicable in the two-image setting. This constraint is designed to promote physically plausible focus transitions by aligning the predicted in-focus likelihood with the focal ordering of the stack. Specifically, it encourages predictions where the in-focus probability decreases smoothly as the focal plane moves farther from the peak, aligning with the physical intuition of focus transition. With only two focal images, such continuity is not well-defined and the constraint becomes less meaningful. Nevertheless, since the model’s performance gains primarily stem from the spatial variational constraint, the limited applicability of the focal constraint is unlikely to significantly affect the model’s overall effectiveness.
> > > >
> > > > If there are any remaining questions or aspects that could benefit from further clarification, we would be happy to address them.

---

> > > > > ### Comment · Reviewer_hJeu · 2025-08-03
> > > > >
> > > > > Thanks for the additional analysis from the authors which has addressed my concerns. My starting point is to explore the trade-off between depth estimation accuracy and device effort when using focal stacks, especially considering the effectiveness of single-image depth estimation recent years. This consideration is important for the practical deployment of such algorithms. Overall, I find this paper meaningful post-rebuttal and am willing to change my score to a weak accept.

---

> > > > > > ### Author Response · Authors · 2025-08-04
> > > > > >
> > > > > > We’re glad that our response was helpful. Thank you for taking the time and effort to review our work.

---

### Official Review · Reviewer_WY53 · 2025-07-01

**Clarity:** 3
**Significance:** 3
**Originality:** 2
**Rating:** 5
**Confidence:** 4

**Summary:**

This paper presents a novel framework for Depth from Focus (DFF) that incorporates two physics-informed variational constraints previously overlooked by existing deep learning-based methods: (1) a spatial variational constraint, and (2) a focal variational constraint.

The spatial variational constraint leverages the physical observation that in-focus regions tend to exhibit coherent and strong gradients, while out-of-focus regions often display blurred or noisy patterns. This constraint is integrated into the network through:

(i) depth gradient supervision with region-dependent weighting (emphasizing in-focus areas), and

(ii) a least-squares operation that implicitly separates in-focus from out-of-focus regions.

The focal variational constraint enforces that the predicted focus probability distribution be smooth and unimodal along the focal sweep. This is implemented via a bidirectional soft monotonicity loss.

Experimental results demonstrate that the proposed framework consistently outperforms state-of-the-art (SOTA) DFF and single-image depth estimation methods. Ablation studies further validate that each proposed component contributes to the overall performance gain.

**Questions:**

1. Can you provide further results to show that the proposed method is better than previous methods when texture-induced artifacts/noises exist?
2. Can you provide further results to show that $\\Gamma$ serves as gradient feature, and $z$ serves as surface feature?
3. Please unify the font style of $z$ in Equation (3) and Figure 1.

**Ethical Concerns:**

["NO or VERY MINOR ethics concerns only"]

**Final Justification:**

The authors provided more detailed explanation on $\Gamma$ and $z$, and also qualitative results to support the claim that the proposed method improves robustness in texture-sensitive regions. And the authors has planned to include these in the final version, making the presentation this paper clearer and easier to follow. Together with the good performance, motivation and novelty of the proposed method, I will suggest a rating of accept.

**Limitations:**

yes

**Quality:**

3

**Strengths And Weaknesses:**

$\\textbf{Strengths}$
+ The integration of the spatial variational constraint and the focal variational constraint into deep learning-based DFF is novel and well-motivated from a physical perspective.

+ The mechanisms proposed for embedding these constraints into the network are well-designed, and the authors have explored a reasonably sized design space to validate their choices.

+ The method achieves strong performance gains over existing approaches, and the ablation studies are comprehensive, covering all proposed components.

$\\textbf{Major Weaknesses}$

+ Throughout the abstract, introduction, methodology, and conclusion, texture-induced artifacts/noises are emphasized as a major cause of degraded performance in prior DFF methods and serve as a key motivation for the proposed framework. However, no targeted experimental results are provided to demonstrate that the proposed method explicitly improves performance in such cases.

+ In line 143, $\\Gamma$ is described as a multi-channel gradient feature, but no visualizations or qualitative results are shown to support this interpretation. Similarly, $z$ is referred to as a latent surface field, yet no qualitative evidence or intuitive explanation is provided to substantiate how this representation reflects surface geometry.

$\\textbf{Minor Weakness}$

The font style of $z$ in Equation (3) and Figure 1 is inconsistent and should be unified for clarity and presentation quality.

---

> ### Author Rebuttal · Authors · 2025-07-28
>
> We thank the reviewer for the valuable feedback and recommendations for improving the manuscript. We list our response to your questions below.
> - - -
> ## 1. Demonstration of Robustness in Texture-Sensitive Regions
> Thank you for highlighting this important point. As correctly noted, texture-induced artifacts and noise often lead to erroneous depth estimates in prior DFF methods, particularly in texture-rich but geometrically flat regions. Our framework addresses this challenge by modeling focus-induced gradient variations, which inherently suppress misleading high-frequency textures and promote structurally consistent depth estimation.
>
> Due to the NeurIPS rebuttal policy this year, we are unfortunately unable to provide additional figures or updated qualitative results beyond the original submission. However, we would like to highlight that several examples in the main paper and supplementary material already help illustrate the robustness of our method to texture-induced artifacts:
>
> - In Figure 4 of the main paper, the first row shows a carpet with repetitive circular patterns on the floor. These texture patterns confuse existing DFF methods, leading to erroneous depth in flat geometry regions. In contrast, our method appears to produce smoother and more consistent depth in this scene.
>
> - The third row in the same figure shows vertical bright dot textures on the inner wall of a brown cabinet. Competing methods often misinterpret these patterns as depth edges, whereas our method yields comparatively more reliable estimates.
>
> - In the fourth row of Figure 4 in the supplementary material, the window composed of small square glass panes contains complex reflections and visual clutter. Prior methods produce inconsistent errors in these regions, but our method regularizes such noise more effectively through the proposed focus-aware gradient-based constraints.
>
> - In the second row of Figure 6 in the supplementary material,  a vertically textured wooden cylinder with strong lighting variations is present. The previous work DFV predicts noisy depth in this area, while ours yields more consistent and clean predictions.
>
> While these examples were included in the original submission, we acknowledge that the associated captions and discussion did not sufficiently emphasize this aspect, and we sincerely thank the reviewer for drawing attention to it.
>
> In the final version, we plan to (i) include more explicit commentary in the captions to draw attention to texture-sensitive regions, and (ii) add a dedicated figure showcasing representative cases of texture-induced errors. We hope these additions will better communicate the practical value of our approach in texture-challenging scenarios.
>
> - - -
> ## 2. On the Geometric Interpretation of $\Gamma$ and $z$
> We appreciate the reviewer’s insightful question regarding the interpretation of $\Gamma$ as a gradient-like representation and $z$ as a surface proxy. While we are limited in providing additional visualizations due to the current rebuttal policy, we would like to clarify that their roles are both mathematically grounded and qualitatively supported by the visualizations already included in the original submission.
>
> **- Theoretical Rationale for Interpreting $\Gamma$ and $z$:**
>
> In our method, each focal stack image is first encoded into a multi-channel directional feature $\Gamma$, which is then used to reconstruct a latent surface field $z$ through a least-squares optimization (Eq. 3 in the main paper). This formulation enforces the constraint that $\nabla z \approx \Gamma$ in the least-squares sense, thereby encouraging $\Gamma$ to represent a spatial gradient field and $z$ to function as the corresponding surface.
>
> To reinforce this structural coupling during training, we supervise the local gradient of $z$ to match the ground-truth depth gradient, specifically in in-focus regions (Eq. 5 in the main paper). The gradient of $z$ here refers to the estimated 2D spatial gradient of the reconstructed surface field, extracted via local convolutional filtering (e.g., a 3×3 kernel), not the learned feature $\Gamma$ itself. As the surface reconstruction from gradients via the least-squares system is differentiable, this supervision ensures that $z$ encodes coherent surface geometry where reliable visual cues exist, while guiding $\Gamma$ to represent localized surface variations consistent with the underlying depth.
>
> **- Focus-Aware Inductive Bias within $\Gamma$ and $z$:**
>
> Before analyzing the visual evidence provided in the paper, we introduce the focus-aware inductive bias embedded in our architecture, which shapes how geometric patterns selectively emerge in $\Gamma$ and $z$.
>
> Our model is designed to encourage the representation to respond selectively to in-focus regions, where reliable visual cues for depth exist. This is achieved through two key mechanisms: (i) focus-dependent integrability, which allows the surface reconstruction to be guided by directional consistency only where valid gradients are present (please refer to lines 162-170 in the main paper), and (ii) focus-weighted supervision, which amplifies training signals in focused areas (Eq. 5 in the main paper). As a result, the network is encouraged to learn gradient-related features that are concentrated and coherent in in-focus regions, while avoiding spurious signals elsewhere. This inductive structure shapes the formation of both the learned gradient field $\Gamma$ and the reconstructed surface $z$ by emphasizing reliable geometric cues in focus regions.
>
> **- Visual Appearance of $\Gamma$:**
>
> We would like to clarify that Figure 2 in the main paper does visualize $\Gamma$, but in a form that aggregates its multi-channel and directional nature into a single composite image. The visualization is constructed by normalizing and combining feature magnitudes to emphasize regions where strong directional responses emerge, thereby illustrating how the network focuses on in-focus areas with reliable gradient cues. While the integrated energy resembles a soft attention map, each directional channel of $\Gamma$ captures localized gradients that collectively contributes to surface reconstruction, guided by the integrability constraint and gradient supervision.
>
> **- Support from Visualized Gradients of $z$:**
>
> In Figure 1 of the supplementary material, we visualize the 2D gradients of the reconstructed surface $z_n$ for each focal plane $n$. These images provide intuitive support for our interpretation of $z$ as a surface proxy: in in-focus regions, the gradient maps tend to exhibit smooth and coherent structures that align with true surface geometry. Conversely, out-of-focus regions produce irregular and noisy gradients, reflecting the model's uncertainty due to unreliable depth cues. This contrast across focus levels serves as an implicit cue that allows the model to infer which regions are in focus and embed this information into its representations, consistent with the intended inductive bias.
>
> We hope these clarifications address the reviewer’s concerns and provide a clearer understanding of our design rationale and its empirical implications. We are also preparing targeted per-channel visualizations and quantitative analyses in the supplementary material, specifically designed to directly reveal how $\Gamma$ captures gradient-like transitions and how $z$ encodes surface-level geometry. These visualizations will serve as concrete evidence supporting our claims and enhance interpretability of the learned representations.
>
>
> - - -
> ## 3. Font Style Consistency
> Thank you for pointing this out. We will ensure the font style of $z$ in Figure 1 is consistent with that in Equation (3) in the final version.

---

> > ### Comment · Reviewer_WY53 · 2025-08-06
> >
> > The authors' response has addressed all my concerns.

---

> > > ### Author Response · Authors · 2025-08-08
> > >
> > > We’re pleased to hear that our response addressed all your concerns, and we appreciate your time in reviewing our work.

---

### Official Review · Reviewer_Gh3D · 2025-07-02

**Clarity:** 3
**Significance:** 3
**Originality:** 3
**Rating:** 5
**Confidence:** 4

**Summary:**

The paper proposes a method to solve the depth from focus problem. The method consists of two main components: handling spatial variational constraints and focal variational constraints, respectively. Spatial variational constraints aim to match the first-order spatial gradient of depth maps to the estimated multi-channel latent representation's spatial gradient. Focal variational constraints try to estimate focus distance to follow the real-world physical behavior---the blur amount monotonically increasing and decreasing from the in-focus depth. Based on two constraints, the depth is estimated by training the focus volume encoder and depth decoder, using the multi-channel latent representation.

**Questions:**

For the rebuttal, I expect the following questions to be answered.

- Are there any other apparent differences between VA-DepthNet, except for the extension for the multi-channel behavior? It seems the confidence weights are removed.
- In comparison against monocular depth estimation methods (categorized as SIDE in the paper), how is the input image selected? It seems the blurred images are used. If so, which focus depth is selected from the focal stack? Are there any criteria for such selection?
- The method seems to work on ideal blur kernel cases. Can this method handle asymmetric blur kernels? Vignetting or optical aberrations cause that asymmetry, which is not rare.

In addition, please indicate the in-focus object in the Figures, if possible, for clarity. (Figure 2, and Figures in the supplementary material)

**Ethical Concerns:**

["NO or VERY MINOR ethics concerns only"]

**Final Justification:**

The authors properly handled my questions and clarified my concerns in rebuttal. Since the authors resolved my concerns, I will keep my ratings.

**Limitations:**

Some potential limitations are raised in questions.

**Paper Formatting Concerns:**

No formatting issues.

**Quality:**

3

**Strengths And Weaknesses:**

The proposed two constraints seem technically sound and valuable. In terms of the spatial variational constraints, although the key idea is from the VA-DepthNet, its extension to multi-channel latent can be shown as the paper's contribution. The focal variational constraints seem to work, based on ablation studies and provided results in the supplementary material. The paper well describes those contributions and also provides sufficient details for experiments. The evaluation result shows state-of-the-art performance. As a result, there is no reason to reject the paper, but with minor considerations that may be answered in the rebuttal.

---

> ### Author Rebuttal · Authors · 2025-07-28
>
> We thank the reviewer for the valuable comments and positive reception of our work. We address your questions as follows:
> - - -
> ## 1. Comparison with VA-DepthNet
> While our method shares a core similarity with VA-DepthNet in leveraging gradient-to-surface reconstruction, it differs in several important aspects both in terms of motivation and implementation.
>
> **- Use of Confidence Weight:**
>
> Our final model also incorporates a confidence weighting mechanism functionally similar to VA-DepthNet. However, we found its effect to be minimal in our setting and therefore chose not to highlight it in the main paper. The table below summarizes the effect of confidence weighting on both models, evaluated on the NYU Depth v2 dataset:
>
> | Method | Setting | RMSE&#8595; | AbsRel&#8595; |
> |-------------|:-----------:|:----:|:------:|
> | VA-DepthNet | w/o confidence weights | 0.316 | 0.089 |
> | VA-DepthNet | w/ confidence weights | 0.304 | 0.086 |
> | Ours | w/o confidence weights | 0.076 | 0.013 |
> | Ours | w/ confidence weights | 0.075 | 0.013 |
>
> In VA-DepthNet, confidence weighting is crucial because the reconstructed surface is used directly as an intermediate depth representation for final refinement. Therefore, suppressing noisy gradients is essential for robust performance.
>
> In contrast, our method is designed to operate on multi-focus image stacks with the primary goal of identifying in-focus regions by analyzing the structural consistency of surfaces across different focal planes. Unlike VA-DepthNet, which relies on accurate surface reconstruction from a single image, our method leverages geometric consistency across focus planes, reducing dependency on precise per-image reconstruction. Additionally, we already suppress unreliable gradients from defocus-blurred regions via a sharpness-based weighting factor $q$, reducing the marginal utility of additional confidence weighting. While confidence weights still offer some benefit in filtering out invalid gradients within sharp regions (e.g., around edges or textured noise), their overall contribution is marginal in our framework, as evidenced by the small quantitative difference.
>
> To prevent potential misunderstanding, we will clarify in the revised version that our model does incorporate confidence weighting, though it plays a limited role in our framework.
>
> **- Lightweight Design for Real-Time Inference:**
>
> While VA-DepthNet aims for high-precision depth estimation from a single image, our model is designed to balance accuracy with real-time inference efficiency for multi-focus depth estimation. This design goal leads to the following key architectural differences:
>
> * We perform surface reconstruction at 1/6 the spatial resolution used in VA-DepthNet. This is feasible in our case because we do not rely on high-fidelity surface recovery from a single image. Instead, we leverage geometric variations across focus planes, which are enriched by the complementary information inherent in the focal stack.
>
> * We use a ResNet-18 encoder and a lightweight decoder designed to fuse information from multiple focus planes. In contrast, VA-DepthNet employs a heavier Swin-Large Transformer and focuses on refining depth from a single image.
>
> As a result, even though our model processes five input images, it remains significantly more efficient in terms of both parameters and runtime:
> | Method | Input Type | #Params (M) | Runtime (s) |
> |-------------------|-----------|:----:|:------:|
> | VA-DepthNet | Single Image | 249 | 0.091 |
> | Ours | 5 Multi-Focus Images | 27 | 0.026 |
>
> - - -
> ## 2. Input Image Selection for SIDE Baselines
> We appreciate this clarification request. For single-image depth estimation (SIDE) baselines, we use the all-in-focus image as input rather than any blurred image from the focal stack. This is because SIDE models are generally trained on all-in-focus data and are not designed to handle defocused inputs. Using blurred images would therefore lead to unfair underperformance. In our evaluation, the NYU Depth v2 and ARKitScenes datasets natively provide all-in-focus RGB images, which were used for SIDE comparisons. We will add this clarification to Section 4.1 to avoid confusion in the final version.
>
>
> - - -
> ## 3. Handling Asymmetric Blur Kernels
> Thank you for bringing up a key aspect that must be addressed for real-world applicability. Our method does not explicitly assume an ideal symmetric blur kernel in its formulation. The spatial variational constraint is designed to capture spatial gradient variations that are consistent with focus-induced sharpness changes, without requiring any parametric blur model. This allows the network to learn focus-dependent structural cues directly from data, regardless of whether the blur is perfectly symmetric or mildly asymmetric.
>
> Similarly, the focal variational constraint encourages the predicted focus probabilities to follow a smooth unimodal pattern across focal planes, which aligns with the physical intuition that sharpness typically peaks at the in-focus depth and decreases away from it. This principle still holds even under mild asymmetric blur common in real-world camera systems.
>
> Our method has been evaluated on a diverse datasets exhibiting different blur characteristics. NYU Depth v2 and ARKitScenes use synthetically generated focal stacks with circular (symmetric) defocus kernels. FOD500 is also synthetic  and assumes idealized blur. In contrast, DDFF12 is a real-world dataset captured with a light-field camera, which naturally introduces moderate blur asymmetry due to lens and sensor characteristics. Our method achieves strong performance across all these datasets, including the real-world DDFF12, indicating its robustness to moderate asymmetry.
>
> Nonetheless, we acknowledge that while our method is generally applicable to scenes with asymmetric blur patterns, its generalization capability may still be limited when trained primarily on symmetric blur and applied to scenes exhibiting severe asymmetry or highly non-Gaussian point spread functions. Performance degradation may occur in such cases due to distributional shifts. This potential limitation is briefly noted in the manuscript (Line 326), and explicit modeling of such effects can be an interesting direction for future work.
>
> - - -
> ## 4. Clarification of In-Focus Regions in Figures
> Thank you for the helpful suggestion. To improve clarity, we will revise Figure 2 and the figures in the supplementary material to indicate the in-focus areas wherever possible in the final version.
>
> Specifically, we plan to use a sharpness weight map $𝑞$ to modulate the RGB input image such that in-focus regions retain their original colors while out-of-focus areas appear visually faded. This will help readers better interpret the focal properties of each input frame and understand how our model responds to focus variation. We will also clarify in the captions that this visualization is solely for interpretability and does not alter the original RGB content.

---

> > ### Comment · Reviewer_Gh3D · 2025-08-05
> >
> > I appreciate the authors for answering my questions and providing clarification. The answers related to the original VA-DepthNet make sense. Adding the contents from the rebuttal will make the paper clearer and understandable. I expect authors to include that additional information from the rebuttal to the paper if accepted.

---

> > > ### Author Response · Authors · 2025-08-05
> > >
> > > We're glad that our clarifications were helpful. The additional information from the rebuttal will be incorporated into the final version of the paper. Thank you for taking the time and effort to review our work.

---

### Note · Authors · 2025-08-12

We sincerely thank all reviewers and the AC for their time and efforts during the review process. We are encouraged that our work was recognized for its innovative contribution, clear motivation, strong performance gains supported by comprehensive experimental validation, and technical soundness. Through our responses, we carefully addressed all questions raised in the initial reviews with detailed clarifications, additional qualitative analyses, and ablation studies. Following the rebuttal and discussion, all reviewers expressed that their concerns were fully addressed.

Unlike prior approaches, our depth-from-focus framework is explicitly built upon the inductive bias of focus-induced gradient patterns that arise uniquely in multi-focus focal stack data. By modeling these patterns with two complementary variational constraints, our method preserves fine geometric edges, suppresses texture-induced artifacts, and maintains physically consistent focus transitions. This physically grounded design plays a central role in achieving the substantial accuracy gains observed.

Across four benchmarks (NYU Depth v2, FoD500, DDFF 12-Scene, and ARKitScenes), our method demonstrates clear improvements over state-of-the-art DFF approaches. These gains are further supported by qualitative analyses, which show robustness in challenging scenes with complex textures, reflections, and low-texture regions. Furthermore, ablation studies confirm the contribution of each component, with the spatial variational constraint proving particularly impactful.

We believe these contributions and findings establish both the novelty and practical value of our approach, especially for applications involving focal stack data, such as computational photography and augmented reality. We look forward to sharing our work with the community.

---

### Decision · Program_Chairs · 2025-09-17

**Decision:**

Accept (poster)

**Comment:**

The paper describes a new technique for depth-from-focus using spatial and focal variational constraints. The spatial constraints are incorporated via gradient-based supervision to prioritize in-focus areas, while the focal constraints are based on a physically plausible unimodal and monotonic distribution. These constraints help to distinguish true depth edges from texture artifacts, leading to SOTA results.

Pros: The combination of spatial and focal constraints is novel. Its effectiveness was demonstrated through extensive experimental validation.

Cons: Validation is based only on indoor scenes, which raises concerns about scalability of method to real-world settings. Multiple focal stack images are required as input as opposed to single-image methods such as DepthAnything.

The main reasons for acceptance are the consensus on novelty and the SOTA results.

The authors provided effective rebuttals to the reviewers’ concerns, which led to three reviewers upgrading their scores. In particular, the authors responded that ground-truth datasets tend to be of indoor scenes only. They also showed that SOTA single-image depth estimation techniques perform more poorly than theirs.